# Promoting favorable interfacial properties in lithium-based batteries using chlorine-rich sulfide inorganic solid-state electrolytes

Dewu Zeng[1], Jingming Yao[1], Long Zhang [1✉], Ruonan Xu[1], Shaojie Wang[2], Xinlin Yan[3], Chuang Yu[4] & Lin Wang[2]

The use of inorganic solid-state electrolytes is considered a viable strategy for developing high-energy Li-based metal batteries. However, suppression of parasitic interfacial reactions and growth of unfavorable Li metal depositions upon cycling are challenging aspects and not yet fully addressed. Here, to better understand these phenomena, we investigate various sulfide inorganic solid electrolytes (SEs), i.e., $Li_{7-x}PS_{6-x}Cl_x$ ($x = 0.6$, 1.0, 1.3, 1.45, and 1.6), via ex situ and in situ physicochemical and electrochemical measurements. We found that the Cl distribution and the cooling process applied during the SE synthesis strongly influence the evolution of the Li|SE interface in terms of microstructure, interphase composition, and morphology. Indeed, for a SE with a moderate chlorine content (i.e., $x = 1.3$) and obtained via a slow cooling process after sintering, the Cl atoms are located on the surface of the SE grains as interconnected LiCl nanoparticles that form an extended LiCl-based framework. This peculiar microstructure facilitates the migration of the Cl ions to the Li|SE interface during electrochemical cycling, thus, favouring the formation of a LiCl-rich interphase layer capable of improving the battery cycling performances.

---

[1] Clean Nano Energy Center, State Key Laboratory of Metastable Materials Science and Technology, Yanshan University, Qinhuangdao, Hebei 066004, China. [2] Center for High Pressure Science (CHiPS), State Key Laboratory of Metastable Materials Science and Technology, Yanshan University, Qinhuangdao, Hebei 066004, China. [3] Institute of Solid State Physics, Vienna University of Technology, Wiedner Hauptstr. 8-10, 1040 Vienna, Austria. [4] State Key Laboratory of Advanced Electromagnetic Engineering and Technology, School of Electrical and Electronic Engineering, Huazhong University of Science and Technology, Wuhan 430000 Hubei, China. ✉email: lzhang@ysu.edu.cn

The rapid growing demand for portable devices, electric vehicles, and large-scale energy storage systems urges lithium batteries with a high energy density as well as high safety[1-3]. Traditional lithium-ion batteries using organic liquid electrolytes, however, cannot meet this requirement in view of the severe safety issue[1]. Instead, all-solid-state lithium batteries (ASSLBs) using solid electrolytes (SEs) are proposed as next generation batteries because of the advantages of SEs including: a high thermal stability to solve the safety issue and working at a broad temperature range, a good mechanical property to retard and suppress Li dendrites, and a decent electrochemical stability window to work with various active materials[4-6].

Albeit these advantages, SEs still cannot realize practical application for energy-dense ASSLBs yet. One of the critical limitations is that Li filaments still deposit/grow inside the SEs even at a low current density (<0.1 mA/cm$^2$) and eventually cause a short circuit upon cycling. Regarding sulfide SEs, furthermore, a long-term (electro)chemical stability of sulfides against either oxide cathode materials or metallic Li needs to be further addressed[7-10]. Mixed ion−electron conducting interphases (MCI) may form at the Li|sulfide interface, which not only decompose continuously the sulfide, but also promote the generation of Li dendrites. As one type of the most attractive sulfide SEs, lithium argyrodites $Li_6PS_5X$ (X = Cl, Br, I) possess a high ionic conductivity, a relative wide electrochemical window, solution-processable syntheses, good deformability, and low-cost raw materials, thus become a highly important family of sulfide SEs[9,11-13]. $Li_6PS_5Cl$ typically shows decent overall performance[13]. It has been found that the degree of $X^-/S^{2-}$ structural disorder and the amount of Li vacancy are the major origins for the different ion transport behavior for $Li_6PS_5X$[14,15]. Among different X elements, the $Cl^-/S^{2-}$ structure is the most disordered with proper lattice softness, leading to a prominent ionic conductivity in $Li_6PS_5Cl$[16-18]. Recently, $Li_{7-a}PS_{6-a}Cl_a$ argyrodites with various Cl contents, especially where $a > 1$[19], have shown interesting behavior that a higher chlorinity leads to stronger $Cl^-/S^{2-}$ site disorders, and moreover, a higher occupancy of Cl atoms at the 4d site of the argyrodite structure, thereby achieving a faster ion transport with an ionic conductivity ~10 mS/cm[20-22]. A high ionic conductivity is necessary for high loading cathode configurations, which is a prerequisite to realize the high energy density.

Although efforts have been already performed to well understand the effect of halogen on the argyrodite-structural disorder and ion transport, to the best of our knowledge, so far, the Li|argyrodite interface is not well understood yet. The Cl-rich $Li_{7-a}PS_{6-a}Cl_a$ argyrodites (e.g., $a = 1.5$) are electrochemically stable against Li, whereas they cannot inhibit Li dendrite penetrations[23]. A multilayer configuration of sulfide SEs is thus designed to remedy the shortcoming[23,24]. Nevertheless, further efforts are required to explore how the chlorinity in argyrodite affects the interfacial compatibility of Li|SE and the Li dendrite suppression capability, and what the impact degree is. Diverse characterization methods should be combined together to answer these questions, for instance, the well-known existing results on the decomposed products at the $Li|Li_6PS_5X$ interface are not resolved yet[25-27]. The products of LiCl, $Li_3P$, and $P_2S_5$ are confirmed for $Li_6PS_5Cl$, but the formed Li–S products are variant based on the observed results of $Li_2S$, $Li_2S_n$, or $S_8$. It is thus very interesting and important to clarify the interfacial performance of the well-engineered $Li_{7-\alpha}PS_{6-\alpha}Cl_\alpha$ toward Li.

Herein, we report that the suppression of unfavorable Li metal depositions (e.g., dendrites) and the (electro)chemical interface stability in Li argyrodites are strongly influenced by the chlorine content (i.e., chlorinity) of the SE. Integrating the in situ measurements with ex situ cryogenic scanning transmission electron microscopy (cryo-STEM) and focus-ion beam scanning electron microscopy (FIB-SEM) measurements, we found that a LiCl shell wraps on an argyrodite core in the case of a high chlorinity (e.g., Cl = 1.3 and 1.6), unraveling the mechanism behind the key role of the chlorinity on the electrochemical performance. The LiCl shells interconnects together to form a microstructure of framework with the localized argyrodite grains encapsulated. The thickness of the shells/frameworks varies with the chlorinity and causes differentiated Li|argyrodite interfaces including the constituents and their quantity, the microstructure, and the distribution. During electrochemical testing, the Cl ions in the frameworks migrate to the Li electrode and re-bond with the local Li ions to reconfigure a dense, even, and uniform LiCl-dominated interphase layer on the surface of the Li electrode, which acts as a self-limiting interphase to avoid the (electro)chemical parasitic reaction of Li|SE. Meanwhile, a moderate chlorinity (Cl = 1.3) with an appropriate thickness (~20 nm) of the LiCl framework is underlined a decisive factor to suppress the Li dendrite growth. The synergistic action of the LiCl framework and LiCl-dominated interphase layer enable ASSLBs using argyrodite SEs with a moderate chlorinity to achieve high electrochemical properties.

## Results

**Structure and ionic conductivity of $Li_{7-x}PS_{6-x}Cl_x$ argyrodites.** The XRD patterns of the as-synthesized $Li_{7-x}PS_{6-x}Cl_x$ ($x = 0.6$, 1.0, 1.3, 1.45, and 1.6, designated respectively as Cl-06, Cl-10, Cl-13, Cl-145, and Cl-16) after annealing are shown in Fig. 1a. All samples are dominated with the argyrodite structure (cubic, space group F -4 3 m), indicating a successful synthesis of the target lithium argyrodite phase. The powder color (Fig. 1b) changes from dark to light with increasing $x$. The halo patterns at low angles (10−20°) are mainly caused by the polyimide film used for preventing the air. A trace of $Li_2S$ and LiCl impurities are observed for the low ($x = 0.6$) and high ($x \geq 1.3$) chlorinity samples, respectively. The LiCl impurity contents (Supplementary Table 1) derived from Rietveld refinements are 2.7, 3.2, and 4.4 vol.% for $x = 1.3$, 1.45, and 1.6, respectively. A peak shift toward the high diffraction angle (magnified at 52.4°) can be observed by increasing $x$, which is originated from the unit cell contraction. The lattice parameter (Fig. 1c) decreases linearly from 9.890 Å for Cl-06 to 9.817 for Cl-13 and then to 9.788 Å for Cl-16, in agreement with the result in literature[28]. The reduction of the lattice parameter by Cl doping on the S sites is considered the combining effects of the smaller ionic radius of Cl compared with that of S, the increased $Cl^-/S^{2-}$ site disorders, and the formation of Li vacancies generated from Cl aliovalent doping. The selected area electron diffraction (SAED) patterns (Fig. 1d and Supplementary Fig. 1) taken from TEM show that Cl-06 and Cl-10 are crystallized, whereas Cl-13 and Cl-16 marginally show a small amount of amorphous phase as indicated by the weak halos. This is against the expectation for some not yet known reasons, as the argyrodite phase should be in a crystalline state after annealing[29].

The Nyquist plots of $Li_{7-x}PS_{6-x}Cl_x$ (Fig. 1e) contain a nearly disappeared semicircle and a spike, which are attributed to contributions from the grain-boundary/bulk and the blocking electrodes[30,31]. The steep linear spike at low frequencies (Supplementary Fig. 2) indicates that the as-synthesized $Li_{7-x}PS_{6-x}Cl_x$ argyrodites are ionic conductors[32]. The incomplete semicircles indicate a small grain boundary resistance[33,34], a typical feature of sulfide-based SEs that is favorable for battery assembly. However, it is hard to distinguish the grain boundary and bulk contribution based on these measured impedance spectra. With the comparable micro-level morphology observed from the SEM images (Supplementary Fig. 3) for the cold-pressed Cl-06 and Cl-13 SE pellets, the shrunken semicircle (Fig. 1e) with increasing Cl concentration ($x$) can be attributed to the reduced

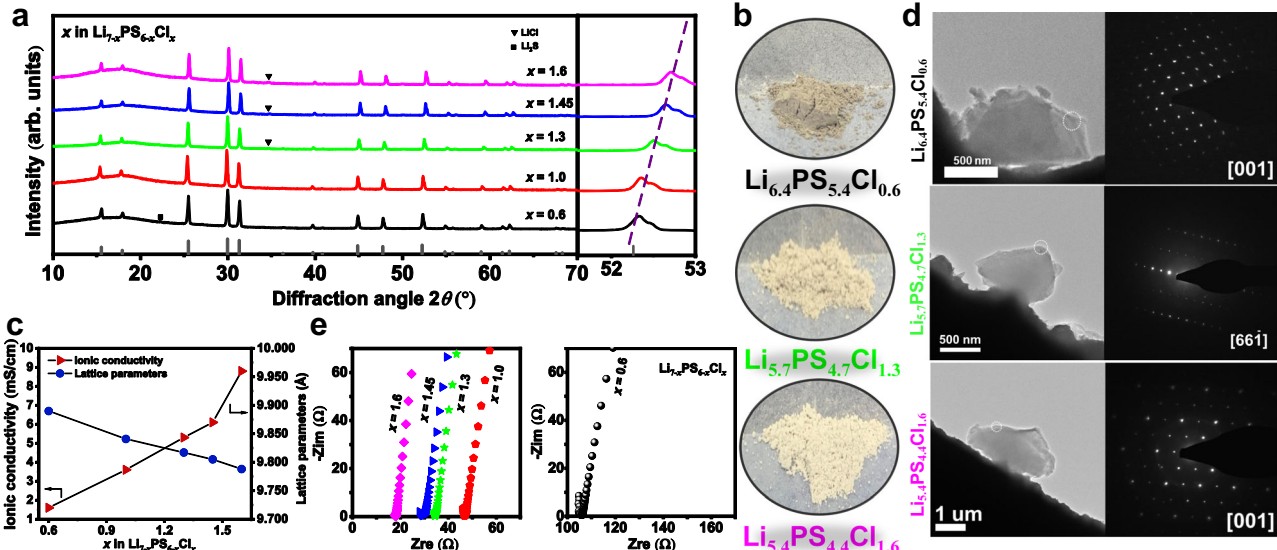

**Fig. 1 Structure and ionic conductivity analyses for $Li_{7-x}PS_{6-x}Cl_x$ argyrodites. a** X-ray diffraction patterns of annealed samples. **b** The color of the powders for selected samples. **c** Chlorinity dependence of the ionic conductivity and lattice parameters. **d** TEM images of Cl-06, Cl-13, Cl-16, and the corresponding SAED patterns. **e** Complex impedance plot of selected frequencies for measurements carried out in In|SE|In cell configuration at 24 °C.

grain-boundary/bulk resistance. Because the grain-boundary/bulk resistance cannot be clearly detected, it is hard to fit the Nyquist plots[31]. The total ionic conductivity is thus calculated from the local minimal resistance at the intersection between the impedance spectrum and the x-axis[35,36]. The value measured at 24 °C (Fig. 1c) increases linearly with increasing Cl concentration until $x = 1.45$, then shows a sharper increase to 8.8 mS/cm (Supplementary Table 1) for $x = 1.6$, which is in accordance with the previous results[21,28].

**Interfacial compatibility of $Li_{7-x}PS_{6-x}Cl_x$ against metallic Li.** To evaluate the effect of the chlorinity on the chemical stability of argyrodites against metallic Li, the time-resolved electrochemical impedance spectroscopy (EIS) was performed intermittently on the $Li|Li_{7-x}PS_{6-x}Cl_x|Li$ symmetric cells with time up to 600 h, as shown in Fig. 2a−d. For the Cl-06 sample, the semicircle of the real part impedance at middle frequencies, relevant to the interface contribution in the range of 1–250 kHz (see the Bode in Fig. 2e), increases intensively to three orders of magnitude compared with the initial value with prolonging the aging time. Nevertheless, when the Cl content changes from a poor to a rich status, the amplification of the semicircles at the same frequency range rapidly goes down. With the highest chlorinity, the semicircle of Cl-16 shows a very limit extension over time. To deeply understand the SE degradation caused by the interfacial degradation, all impedance spectra were fitted by using an equivalent circuit of $R_b(R_{gb}\text{-}CPE_{gb})(R_{int}\text{-}CPE_{int})W$, which consists of a bulk resistance ($R_b$), a parallel unit ($R_{gb}\text{-}CPE_{gb}$) related to the grain boundary resistance of the argyrodite SE, a second parallel unit ($R_{int}\text{-}CPE_{int}$) related to the interfacial layer resistance, and a Warburg element ($W$)[37], as shown in the inset of Fig. 2e. The Bode diagram (Fig. 2e) displays that $R_{int}$ locates in the frequency range of 1–250 kHz, which is in line with the literature data[38]. The $R_{int}$ values and corresponding errors acquired by fitting the impedance plots shown in Fig. 2a−d are listed in Supplementary Table 2. The time evolution of $R_{int}$ as a function of Cl content is shown in Fig. 2f. The Cl-poor Cl-06 sample shows a severely enlarged $R_{int}$ upon aging time compared with the Cl-rich samples. The $R_{int}$ value increases from an initial value of 209–591 Ω at 100 h, and then rapidly increases to 9093 Ω at 600 h; while those

of Cl-13 and Cl-16 increase slowly from 50 and 19 Ω, respectively, to 486 and 256 Ω at 6000 h. This indicates a suppressed chemical reactivity of argyrodite toward Li by increasing chlorinity.

The critical current density (CCD) is an important parameter to evaluate the ability of SE to sustain the maximal current density without a short circuit by Li dendrites[39,40]. It was tested on $Li|Li_{7-x}PS_{6-x}Cl_x|Li$ symmetric cells at 24 ± 4 °C with both initial and step-increased current densities of 35 μA/cm². The voltage-time plots for $Li_{7-x}PS_{6-x}Cl_x$ with arrow-marked short circuit points are displayed in Supplementary Fig. 4. To ensure the reliability of the CCD data, the repeatability tests using three different production batches were performed under the same conditions for $Li_{7-x}PS_{6-x}Cl_x$ ($x = 0.6$, 1.0, 1.3, and 1.6). Figure 3a shows the individual data points together with the mean values and error bar (presented by standard deviation). Although the CCDs (Fig. 3a) display a slightly scattered distribution within the individual composition, the variation trend shows that the CCD gradually rises with increasing the chlorinity, reaches a maximum value over 1 mA/cm² for $x = 1.3$, and then quickly goes down with no further increase. It is surprising that, despite of the highest ionic conductivity, Cl-16 sustains the lowest CCD (0.56 mA/cm²), even lower than that of Cl-06. A moderate Cl concentration is thus proposed to be better for suppressing Li dendrites.

The long-term Li plating/stripping reversibility was tested on the Li|SE|Li symmetric cells for $Li_{7-x}PS_{6-x}Cl_x$ ($x = 0.6$, 1.0, 1.3, and 1.6) SEs under various current densities at 24 ± 4 °C. This performance depends on Li deposition/growth and Li|SE interfacial degradation, thereby giving insight into the overall interfacial compatibility of $Li|Li_{7-x}PS_{6-x}Cl_x$. We first use a moderate current density of 0.25 mA/cm² (Fig. 3b−e) to ensure a long-term cycling for all cells. Though the polarization voltage enlarges upon cycling due to the formation of interphases at Li|SE interface, the increment rate strongly varies with the chlorinity. The argyrodite degradation is severe in Cl-06 (Fig. 3b), especially after 200 cycles the voltage rapidly increases. On the contrary, the degradation is mild in Cl-10 (Fig. 3c) and Cl-13 (Fig. 3d), and is negligible in Cl-16 (Fig. 3e) even at the 400th cycle, as one can see that the voltage curve is almost flat in Cl-16. This scenario confirms that detrimental interfacial reactions occur in the

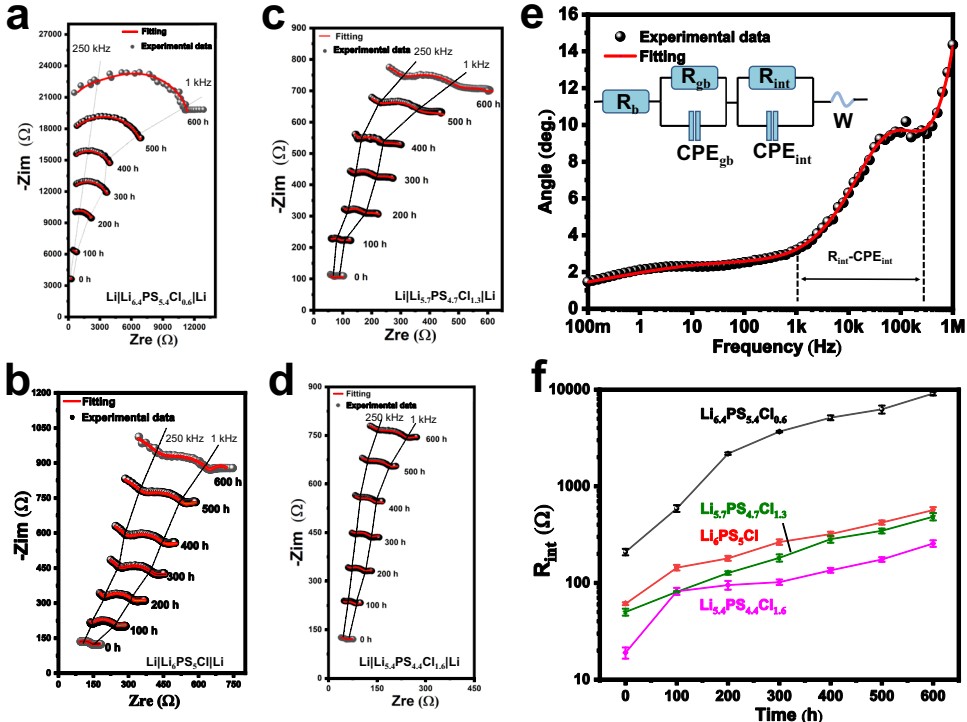

**Fig. 2 Time-resolved EIS tested intermittently on the Li|Li$_{7-x}$PS$_{6-x}$Cl$_x$|Li symmetric cells for 600 h at 24 °C.** Evolution of the impedance signal in Nyquist plots for the samples of Cl-06 (**a**), Cl-10 (**b**), Cl-13 (**c**), Cl-16 (**d**). **e** Representation of the impedance signal (Cl-13 at 300 h) in a Bode plot. The inset displays the equivalent circuit diagram adopted for fitting the Bode plot and the impedance spectra (**a–d**). **f** Interfacial layer resistance of Li|SE as a function of storage time.

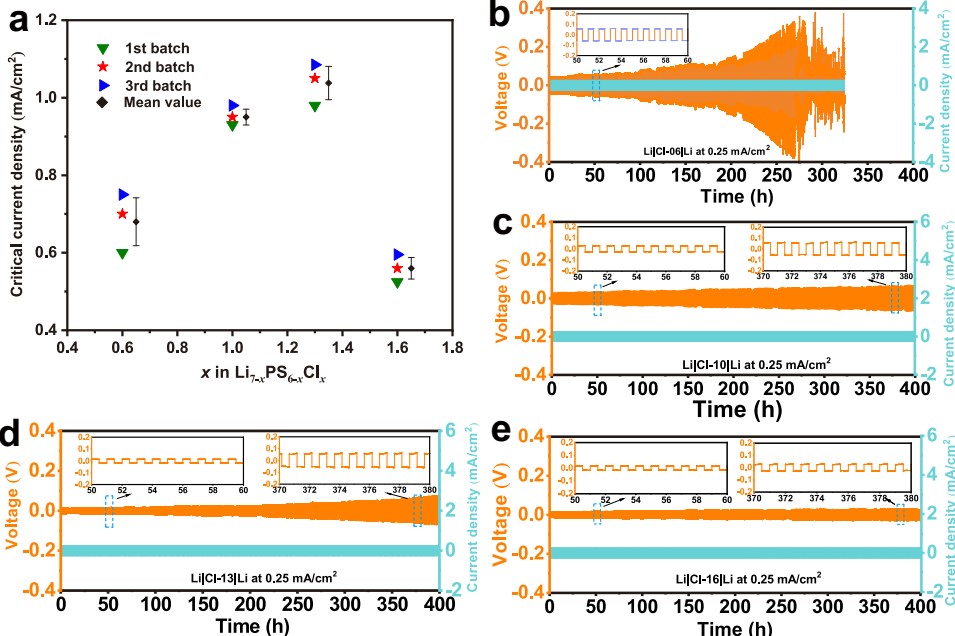

**Fig. 3 Li dendrite suppression capability and long-term interface stability of Li|SE|Li using Li$_{7-x}$PS$_{6-x}$Cl$_x$ (x = 0.6, 1.0, 1.3, and 1.6) SEs at 24 ± 4 °C.** **a** Critical current density (CCD) for different argyrodite batches. Long-term Li plating/stripping cycling at 0.25 mA/cm$^2$ for Cl-06 (**b**), Cl-10 (**c**), Cl-13 (**d**), and Cl-16 (**e**).

Cl-poor SEs (e.g., Cl-06), and a high chlorinity especially for Cl > 1.6 is favorable to restrain parasitic reactions in Li|argyrodite. The cell using the Cl-06 SE suffers a short circuit at the 270th cycle. In contrast, no short circuit is observed in the cells using the other SEs for x ≥ 1.0.

In situ EIS was performed on the cells subjected to long-term Li plating/stripping (Fig. 3b−e) for every hundred cycles, as illustrated in Fig. 4. According to the bode plot (Supplementary Fig. 5), the interface region of Li|SE locates in the frequency range of 1–200 kHz. The impedance spectra were fitted by using an

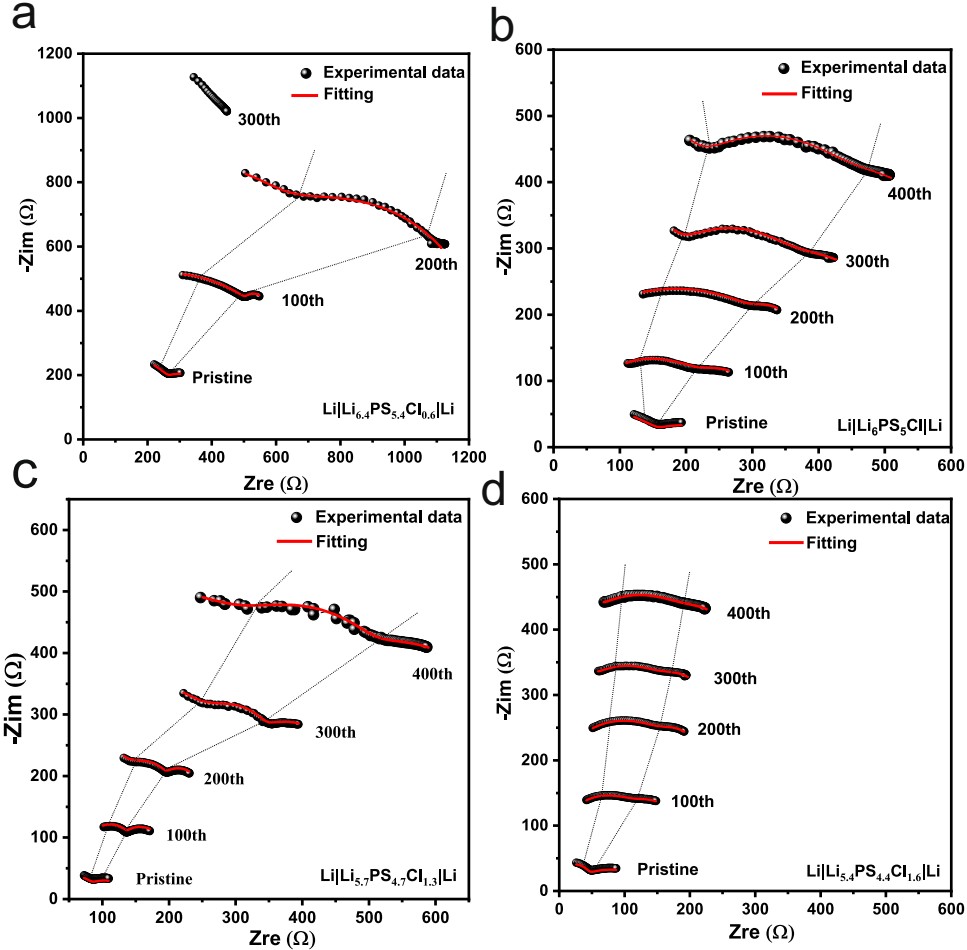

**Fig. 4 Interfacial evolution of the Li||Li cells subjected to Li plating/stripping (Fig. 3).** The impedance spectra were acquired for every 100 cycles from the in situ EIS measurements on the cells using Cl-06 (**a**), Cl-10 (**b**), Cl-13 (**c**), and Cl-16 (**d**) SEs.

equivalent circuit of $R_b(R_{gb}\text{-}CPE_{gb})(R_{int}\text{-}CPE_{int})W$ (as mentioned above). The $R_{int}$ values and corresponding errors are listed in Supplementary Table 3. The Cl-06 and Cl-10 samples show a continuously and intensively enlarged interfacial resistance. The resistance of Cl-13 increases slowly within 300 cycles (≤92 Ω) but rapidly after that, reaching a value of 178 Ω at the 400th cycle, which, however, is still much lower than those of Cl-06 and Cl-10 (after the 200th cycle). It is notable that in Cl-16 the increase of the interfacial resistance becomes even slower after 200 cycles, thus Cl-16 shows a much smaller resistance than the other cells over the whole measured cycle range. The interfacial resistance of Cl-16 at the 400th cycle (49 Ω) does not significantly increase compared with the initial value (17 Ω), indicating an improved stability against Li.

The distinction of cycling performance upon chlorinity is enlarged by forcing a high current density of 0.5 mA/cm², as shown in Fig. 5. The slight fluctuation of the voltage (lower inset of Fig. 5a) is induced by the variation at ambient temperature (24 ± 4 °C) during the long-term cycling. Cl-13 (Fig. 5a) remains a good cycling stability with flat voltage platforms (upper inset of Fig. 5a). No short circuit is observed in Cl-13 for over 1000 cycles, whereas a short circuit occurs on the other chlorinities (see Fig. 5b–d) for less than 30 cycles, confirming that Cl-13 has a far better Li dendrite suppression capability than the other compositions. Along with the time-resolved EIS results, we conclude that Cl-13 with a moderate chlorinity possesses an (electro)chemical interface stability toward Li and a better Li dendrite suppression capability.

**Charging/discharging performance of the cells using Li and Li-In electrodes.** The cell performance is not only dependent on the ionic conductivity of SEs but also on the interfacial stability[36,41]. To further verify the compatibility of argyrodite against Li, ASSLBs were assembled using different negative electrodes including Li and Li−In. In order to exclude the influence of the cathode materials, LiNbO₃-coated $LiNi_{0.6}Mn_{0.2}Co_{0.2}O_2$ (LNO@NCM622) and $LiNi_{0.8}Mn_{0.1}Co_{0.1}O_2$ (LNO@NCM811) were used as active materials and mixed respectively with the commercial $Li_{9.54}Si_{1.74}P_{1.44}S_{11.7}Cl_{0.3}$ (LSPSC) as ion conductors in the positive electrode[42–44].

Figure 6a−d shows the electrochemical performance of Li|SE| LNO@NCM622 using Cl-13 and Cl-06 cycled at 0.1 mA/cm² at 24 ± 4 °C. The former demonstrates significantly enhanced performance than the latter. Firstly, the initial Coulombic efficiency of the former (68%) is higher than that of the latter (55%). Secondly, the Coulombic efficiency of the former keeps constant and approaches 100% after the first few cycles, while that of the latter is fluctuant in the range of 80–100%. Thirdly, the former shows a good cycling stability with specific capacities of ~137 mAh/g, while the capacity of the latter rapidly decreases upon cycling. The charging-discharging profiles of the cell using Cl-13 (Fig. 6c) show low polarization voltages and are well overlapped, indicating a good interfacial compatibility between the SE and both electrodes. In a sharp contrast, intensively increased polarization voltages and capacity decays are observed for the cell using Cl-06 (Fig. 6d), particularly during the

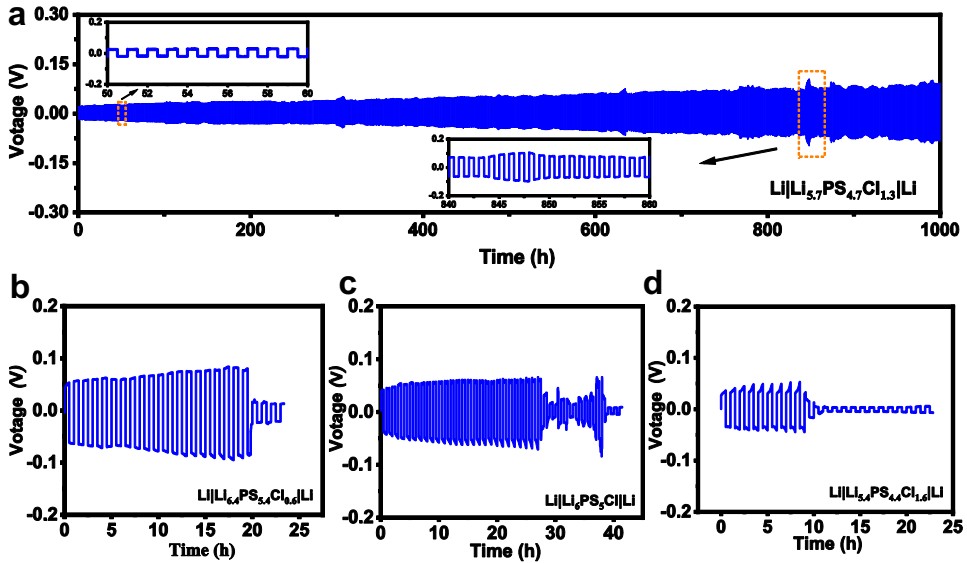

**Fig. 5 Symmetric Li||Li cells cycling performance for SEs with different chlorine content.** Li plating/stripping cycling on Li|SE|Li symmetric cells at 0.5 mA/cm$^2$ at 24 ± 4 °C using Cl-13 (**a**), Cl-06 (**b**), Cl-10 (**c**), and Cl-16 (**d**) SEs.

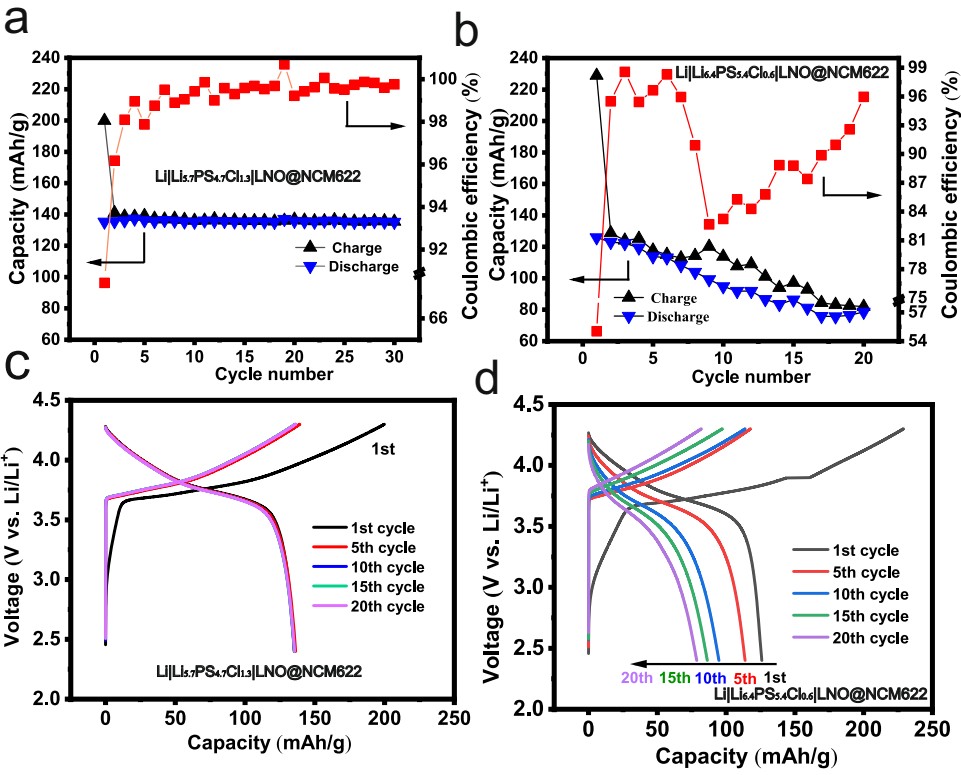

**Fig. 6 Electrochemical cycling performance of Li|SE|LNO@NCM622 cells with Cl-06 and Cl-13 SEs.** Capacity of Li||LNO@NCM622 cells for the first 30 cycles run at 0.1 mA/cm$^2$ at 24 ± 4 °C using Cl-06 (**a**) and Cl-13 (**b**) SEs, and the corresponding charge/discharge curves (**c**, **d**).

discharging process. This confirms reactions at Li|SE side in Cl-06. Seen from the first charging profile for both cells, there exists an additional oxidative process presenting as a slope below 3.7 V. Generally, such an oxidative process is regarded as side effects at the cathode including:[8,45] the formation of a space-charge layer between the sulfide ion conductor and the active material, and a volume change by Li$^+$ storage. However, one should note that Cl-06 shows a gentler slope for the first charging profile than Cl-13. Considering that the two cells have the same assembling conditions and same components except for the electrolyte part, the different slopes can only be originated from the different chlorinities. That is, in addition to the impact of the positive electrode, the initial charging profile is also affected by the Li|SE interface. We speculate that the formed interphases at the Li|Cl-06 interface alters the local voltage difference and thus impacts the Li$^+$ storage process.

The Li|SE interface can also be investigated by replacing the metallic Li with a Li–In alloy (Li: 5 wt%) electrode. Fig. 7a shows

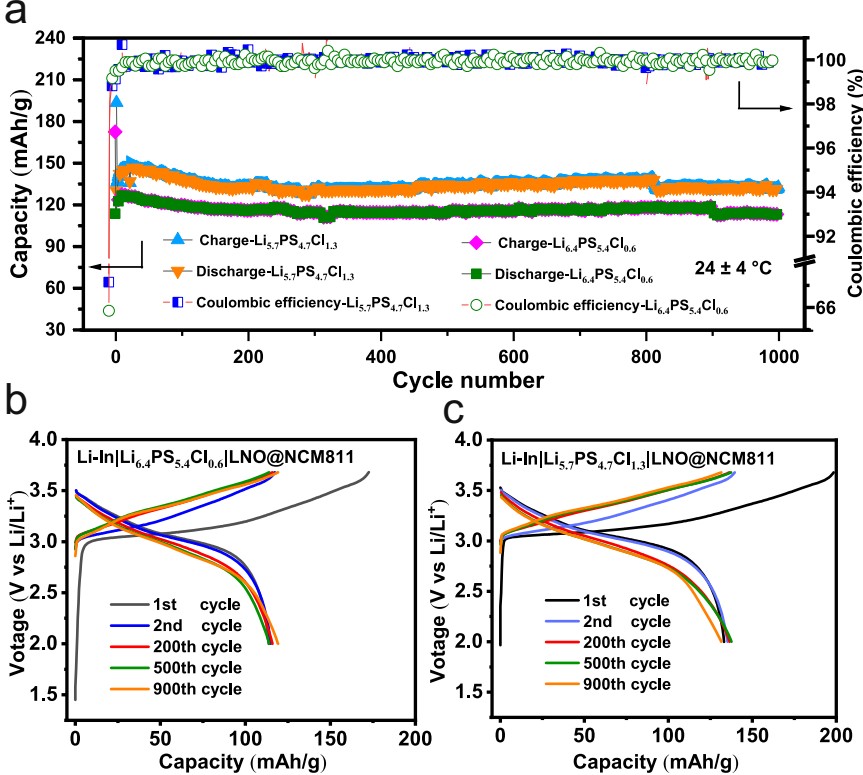

**Fig. 7 Electrochemical cycling performance of Li-In||LNO@NCM811 cells. a** Capacity and Coulombic efficiency for 1000 cycles run at 0.5 mA/cm$^2$ at 24 ± 4 °C using Cl-06 and Cl-13 SEs. Charge/discharge profiles for Cl-06 (**b**) and Cl-13 (**c**) SEs.

the cycling stability of the Li-In|SE|LNO@NCM811 at 0.5 mA/cm$^2$ at 24 ± 4 °C. The two cells using Cl-06 and Cl-13 SEs show comparable initial Coulombic efficiencies around 66% and similar charging–discharging profiles (Fig. 7b, c), confirming that the oxidation step below 3.7 V is triggered by the unstable feature of Cl-06 against Li. The cells using both SEs demonstrate good cycling stabilities over 1000 cycles at 0.5 mA/cm$^2$. However, the specific capacity of Cl-13 (~148 mAh/g@50th cycle) is higher than that of Cl-06 (~123 mAh/g@50th cycle) and is comparable to that of Cl-16 (~145 mAh/g@50th cycle, Supplementary Fig. 6). This is because Cl-06 consumes more active Li than Cl-13 and Cl-16 at the Li|SE interface, further confirming that the Cl-rich argyrodites are favorable for a stable Li|SE interface.

**Physicochemical characterizations of the Li argyrodite SEs.** The collected data in this work present the correlation between the chlorinity and electrochemical performance in Li$_{7-x}$PS$_{6-x}$Cl$_x$. First, a middle chlorinity (Cl-13) exhibits the highest CCD, while the highest chlorinity (Cl-16) shows the lowest CCD, despite of its highest ionic conductivity. Second, a higher chlorinity forms a self-limiting interface and thus gains better compatibility against Li. Third, a higher chlorinity limits the undesired oxidative step occurred below 3.7 V at the initial charging process, indicating that the rapidly formed Li|SE interface also affects the Li$^+$ storage process in addition to the side effects occurred inner the cathode. Additionally, a higher chlorinity leads to the appearance of an amorphous phase on the argyrodites. To unravel these scenarios, cryo-HRTEM along with ex situ SEM and X-ray photoelectron spectroscopy (XPS) as well as in situ Raman spectroscopy were performed to get a deeper insight on the mechanism behind.

The interface regions of the Li|Li$_{7-x}$PS$_{6-x}$Cl$_x$|Li ($x$ = 0.6, 1.3) symmetric cells after Li plating/stripping for 200 cycles (Supplementary Fig. 7) were collected for XPS analysis. The

specimens were prepared and transferred in Ar atmosphere to prevent oxidation. The XPS maps of the pristine Cl-06 (Supplementary Fig. 8) and Cl-13 (Fig. 8a) are comparable. The peak positions of S 2$p$ at 161.4 and 162.5 eV and those of P 2$p$ at 131.8 and 132.6 eV are attributed to PS$_4$$^{3-}$. No impurity peak is observed in Cl-13. Comparably, Cl-06 shows impurity phases that increase after cycling. The S 2$p$ spectrum of Cl-06 (Fig. 8b) represents S-S and Li$_2$S impurity peaks. The S-S peaks are determined to be Li$_2$S$_n$[46,47]. The P 2$p$ spectrum shows that except for the PS$_4$$^{3-}$ peak, there also exist peaks of reduced phosphorus species and Li$_3$P. In some studies[8], the absence of Li$_3$P may because the cycle number is insufficient to form adequate Li$_3$P. This analysis reveals that Cl-06 is not only decomposed to the commonly observed Li$_2$S, P-based phases, and Li$_3$P[48,49], but also to the rarely observed Li$_2$S$_n$. However, Li$_2$S$_n$ is not detected for the cycled Cl-13 (Fig. 8c), in which only the commonly reported Li$_2$S is observed. This indicates that the formation of Li$_2$S$_n$ is strongly dependent on the chlorinity in the argyrodite SEs. Using the semi-quantitative analysis (Fig. 8d), we can estimate that the remaining PS$_4$$^{3-}$ proportion of Cl-13 is 81 at% but that of Cl-06 is only 48 at% (based on P 2$p$ spectra). Meanwhile, the amounts of all decomposition products of Cl-13 are much lower than those of Cl-06, confirming that the argyrodite degradation is intensively mitigated by the high chlorinity.

As the decomposition products such as Li$_2$S$_n$ and the degree of decomposition at Li|SE interface are closely related to the chlorinity in argyrodite, the interfacial microstructure of Li|SE should be varied with the chlorinity and is worth being carefully investigated. The morphology of the Li|SE interface is thus taken from the Li|SE|Li symmetric cells after Li plating/stripping tests. In order to get sufficient and reliable information, the cells after the long-term cycling without short circuits are adopted. The cell using the Cl-13 SE was run for 400 cycles (Fig. 3d), and that using Cl-06 was run for 200 cycles (Supplementary Fig. 7a) to avoid a

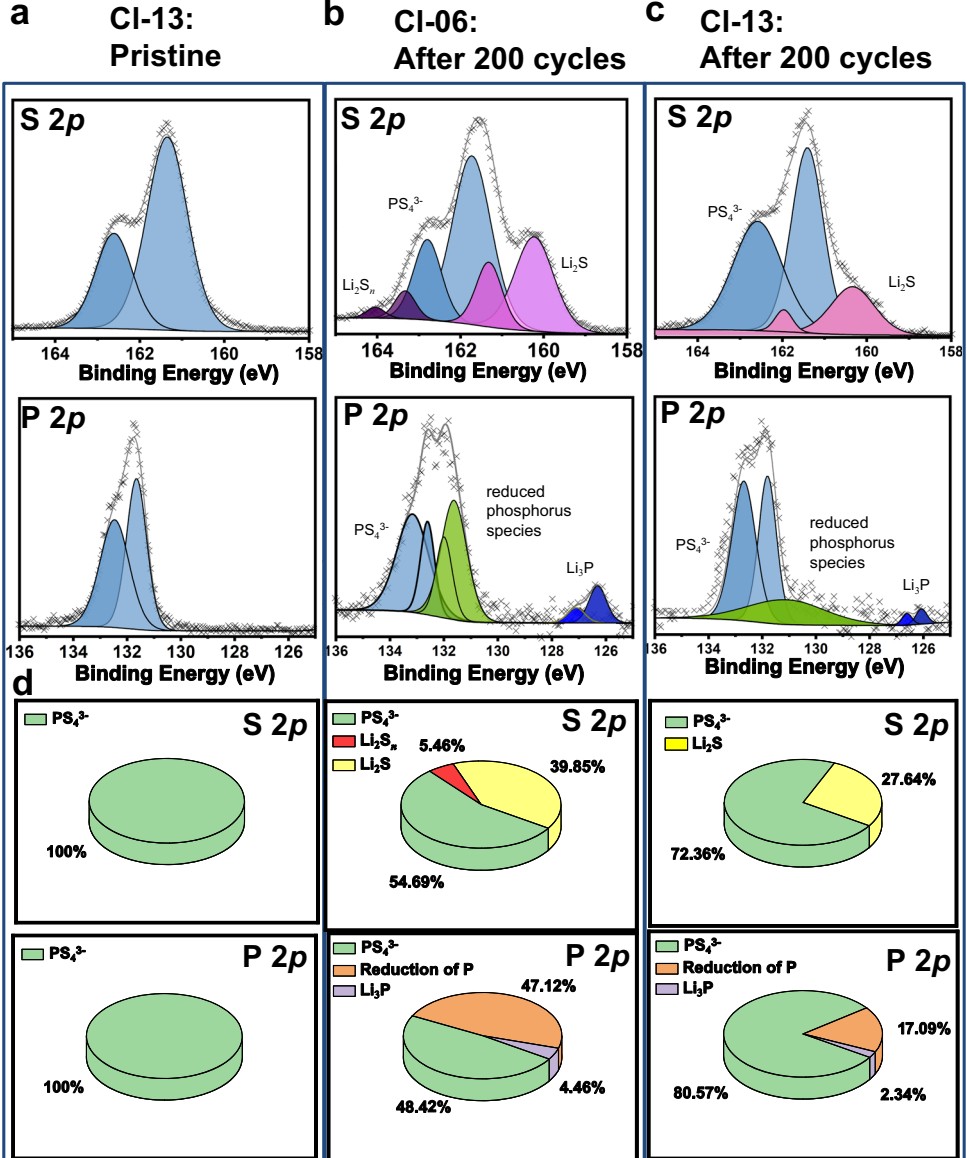

**Fig. 8 XPS evaluation on Li|SE interface for Li||Li symmetric cells using Cl-06 and Cl-13 SEs after Li plating/stripping for 200 cycles. a** Cl-13 before cycling. Cl-06 (**b**) and Cl-13 (**c**) after 200 cycles. **d** semi-quantitative analyses of the interphases.

short circuit. The SEM and EDS images for the surface and the cross section at the interface are displayed in Fig. 9. The SEM images for the surface were captured from the fractured surface inside the interface rather than next to the Li electrode, because the interface layer and the Li electrode are stuck together and thus hard to be separated. Seen from the EDS mapping of the surface (Fig. 9a and Supplementary Fig. 9a), the decomposition products in the Cl-06 cell are distributed as aggregated morphology, which can be seen from the cross-sectional view (Fig. 9b). Typically, the $P_2S_5$ aggregations with a width over 10 μm connect together. In contrast, Cl-13 (Fig. 9c and Supplementary Fig. 9b) displays homogeneously distributed decomposition products without agglomeration. Notably, a LiCl-dominated interphase layer is found next to the Li electrode. Considering that this LiCl layer comes from the LiCl nanoshells (see below) and a large proportion of $PS_4^{3-}$ exists in the Li|SE interface region (Fig. 8), this LiCl layer should contain argyrodite and/or $Li_3PS_4$ in addition to LiCl, i.e., a LiCl-dominated interphase layer mixed

with Li superionic conductor. This layer is smooth and homogeneous with a thickness of $55 \pm 6$ μm, where the particles must be small because no agglomeration is observed. It has been reported that a solid electrolyte interphase (SEI) with a nanosized LiCl mixture possesses fast charge-transfer kinetics[50,51]. From the SEM image (Fig. 9c), we know that this is a high quality LiCl-dominated interphase layer as it is dense, even, and uniform, which is favorable to gain a low resistance SEI[52]. Therefore, even though this interphase layer is as thick as $55 \pm 6$ μm, it still allows fast ion transport kinetics. LiCl is also an electron insulator, so this LiCl-dominated interphase layer is a good buffer layer to stabilize the Li|SE interface[53]. On the contrary, the LiCl layer in Cl-06 (Fig. 9b) is irregular, showing a rough edge and uneven thicknesses in a range from few to 50 μm. This well interprets why the Cl-rich argyrodite SEs can easily keep constant and low overpotentials (Fig. 3d, e and Fig. 5a) during the Li plating/stripping process, i.e., via forming a self-limiting interface. Conversely, the LiCl layer in the Cl-poor argyrodites cannot

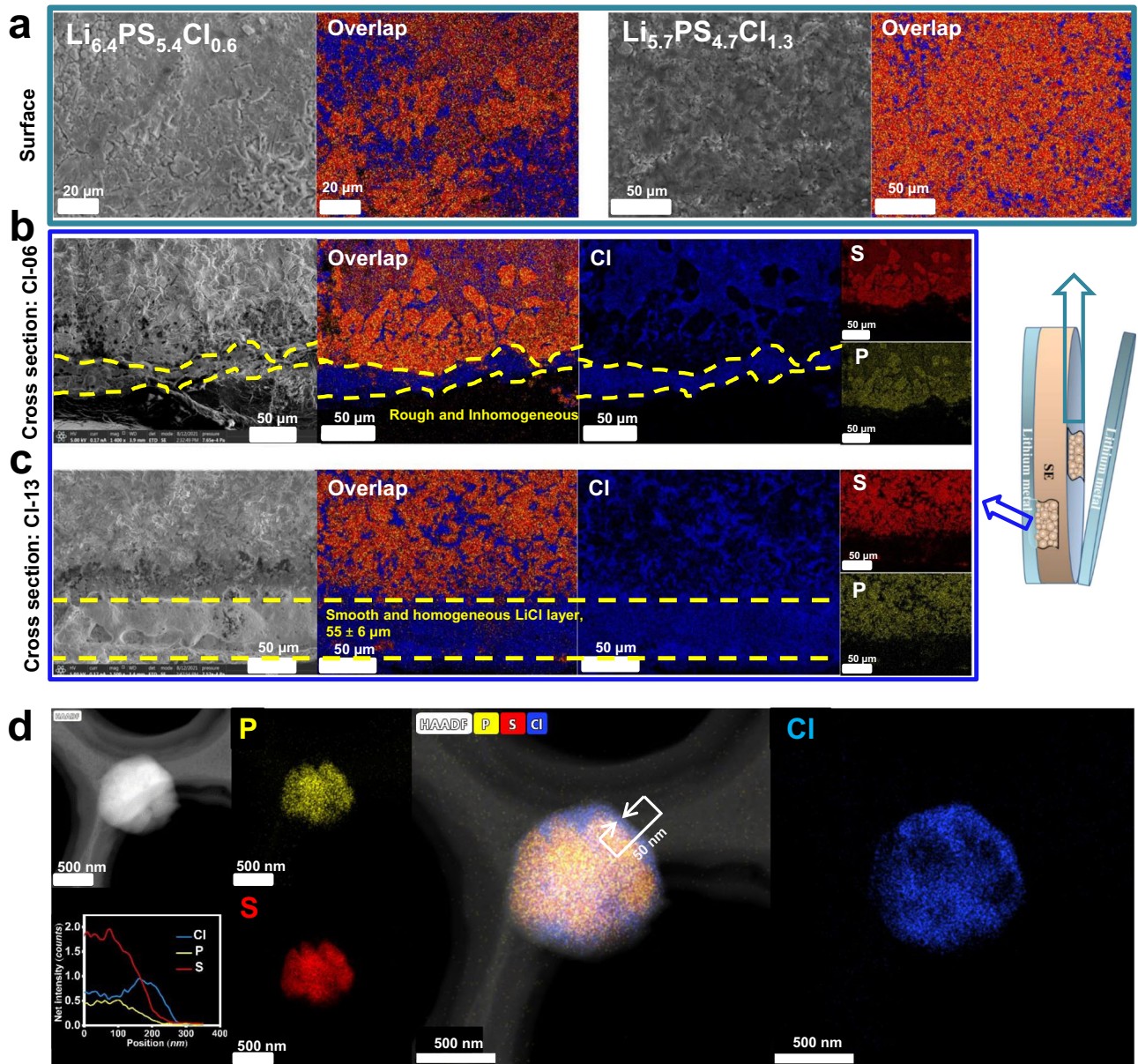

**Fig. 9 Influence of chlorinity on microstructure. a–c** SEM and EDS analyses on the Li|SE interface for Li||Li symmetric cells after plating/stripping cycling using Cl-06 (200 cycles) and Cl-13 (400 cycles) SEs. **a** SEM and EDS images on fracture surface. Cross-sectional SEM and EDS images for Cl-06 (**b**) and Cl-13 (**c**). **d** Cryogenic STEM-HAADF and EDS images of as-prepared Cl-16 powder.

obstruct the continuous decomposition of the argyrodite against Li. Moreover, $Li_2S_n$ induced by a low chlorinity has a higher electronic conductivity but a lower ionic conductivity than $Li_2S$[54]. The irregular LiCl-dominated interphase layer and the electron-conducting $Li_2S_n$ combined together lead to a continuous and severe degradation on the argyrodite SEs, which intensively increases the amount of agglomerated $P_2S_5$ as well as the electron-conducting $Li_2S_n$ and $Li_3P$, eventually forms MCI. This is the reason why the Cl-poor argyrodite SEs are severely decomposed by Li.

Nonetheless, the mechanism for the formation of the differentiated interfaces and thick LiCl-dominated interphase layer is not clear yet. In view of the XRD data that the argyrodite phase is dominated, one would wonder where the LiCl-dominated interphase layer comes from and, more importantly, why they form diverse morphologies. We thus try to figure out how Cl is distributed in argyrodite with regard to chlorinity.

Cryo-STEM integrated with EDS was employed to evaluate the as-prepared Cl-06, Cl-13, and Cl-16 powders. The specimens were held at cryogenic temperature (−170 °C) throughout the preparation and imaging processes to protect the argyrodites from damaging by the air and the imaging beam. The cryo-STEM-HAADF images and corresponding EDS mappings for P, S, Cl, and their overlaps are illustrated in Fig. 9d and Supplementary Fig. 10. To avoid any reaction between the argyrodites and solvent, the powders were spread onto the STEM grid, and then the grid was purged with Ar gas[55]. No liquid was used in the preparation process. The observed single powder is composed by several aggregated grains. Combining the overlaps and the individual elemental mapping images, it is surprising to find that, unlike uniformly distributed P and S, the Cl atoms in Cl-16 (Fig. 9d) primarily exist on the surface of each grain by forming a nanoscale shell layer, which connects with each other and constructs a framework. Only a part of Cl atoms locates

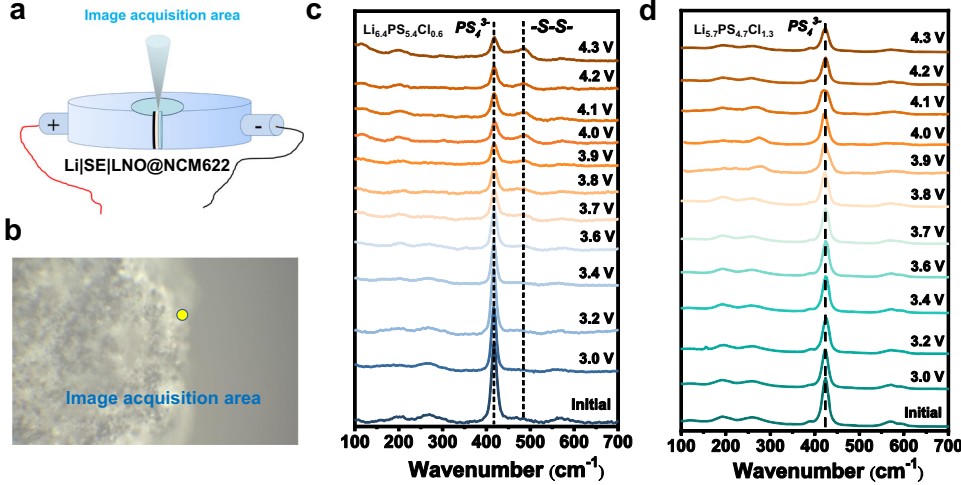

**Fig. 10 In situ Raman spectroscopy detection at the Li|SE interface region on Li|SE|LNO@NCM622 cells with Cl-06 and Cl-13 SEs. a** Drawing of the in situ Raman mold. **b** Image of laser focusing area on the Li|SE interface. In situ Raman spectra of Cl-06 (**c**) and Cl-13 (**d**) in charging process run under 0.1 mA/cm² at 24 °C.

inside the argyrodite lattice to implement a successful doping. Similar elemental distribution behavior is also observed in Cl-13 (Supplementary Fig. 10b), except that the nanoshell in Cl-13 is thinner than that in Cl-16. From the cryo-STEM-EDS line scan, we estimated the thickness of the shell is about 50 nm for Cl-16 and 20 nm for Cl-13. Because the Li element cannot be detected and no other element is observed by EDS, the Cl-contained framework is considered to be LiCl. SEM-EDS was carried out on a cold-pressed pellet of the pristine Cl-13 over 7 random regions under a 20,000× magnification. The selected images are displayed in Supplementary Fig. 11. Except for one region that displays an agglomerated LiCl particle (~2 μm), the other regions display a homogeneous Cl-distribution, indicating that the proportion of the agglomerated LiCl particles is very low.

This observation of LiCl nanoshells is reproducible in other powders, where the representative images taken from Cl-16 are illustrated in Supplementary Fig. 12. However, this scenario is not observed in Cl-06. As illustrated in Supplementary Fig. 10a, Cl shows a homogeneous distribution, and no LiCl nanoshell is observed, indicating that all Cl atoms are successfully substituted on the S sites. In addition to the chlorinity, the sintering technique is also a key prerequisite to form the LiCl nanoshell. To evaluate the influence of the heat treatment, two different cooling processes were adopted after sintering via: (1) cooling the sample to 400 °C slowly and then quenching in the ice water, and (2) directly quenching in the ice water. As shown in Supplementary Fig. 13, there is no LiCl nanoshell for the directly quenched sample, indicating that a slow cooling process at the high temperature regions is a crucial factor. In view of the sintering effect, the LiCl nanoshells could be generated through in situ precipitating from the $Li_{7-x}PS_{6-x}Cl_x$ solid solution, otherwise, LiCl would exist as agglomerated particles rather than homogeneous coating. Consequently, a high chlorinity and a proper sintering process are believed to be the key points for the Li argyrodite to form a structure with localized Li argyrodite grains wrapped in the LiCl framework.

## Discussion

LiCl is not an ideal ion conductor[53], so the LiCl framework would hinder the interparticle ion transport, and deteriorate the ionic conductivity in the high chlorinity argyrodites. However, an inverse correlation is seen with a gradually reduced total resistance and an increased ionic conductivity (Fig. 1c) upon

increasing the chlorinity. The major reason is that the degree of $X^-/S^{2-}$ structural disorder and the Li vacancy concentration increase with increasing chlorinity, which enhances the ionic conductivity. Meanwhile, the LiCl nanoshells are thin enough (dozens of nanometers in thickness) to allow a fast Li⁺ transport[51]. In addition, we speculate that a space-charge layer exists between the argyrodite and the LiCl nanoshell, which also contributes to an increased ionic conductivity. The space charge effect widely exists in composites such as the conductor–insulator and conductor–conductor systems[56,57]. In this, a high conduction matrix is the crucial factor for the high overall conductivity of a composite[58]. The Cl-rich argyrodites with a high ionic conductivity due to the enhanced lattice disorder and Li vacancies belong thus to such a proper matrix. Therefore, the LiCl nanoshell does not degrade the ion transport of the argyrodite-LiCl composite SEs.

The existence of the LiCl shells/frameworks well addresses the mechanism on the formation of the even, uniform, and dense LiCl-dominated interphase layer at the Li|Cl-13 interface (Fig. 9c). As the bandgap of LiCl is 6.3 eV and that of $Li_6PS_5Cl$ is 2.2 eV[59,60], LiCl is a good electronic insulator to restrain the Li⁺ reduction on the surface[53]. The electronic conductivity of as-synthezied $Li_{7-x}PS_{6-x}Cl_x$ (Supplementary Table 1) decreases with increasing chlorinity. Additionally, LiCl is compatible with metallic Li[14]. For the Cl-rich argyrodites, the shell-interconnected LiCl frameworks encapsulating the argyrodite grains play critical roles on improving performance. At the beginning of the chemically contacting and electrochemical cycling, the LiCl nanoshells act as initial protections by inhibiting the parasitic reactions between the argyrodite SE and Li. During the further electrochemical cycling, the Cl ions in the nanoshells migrate to the surface of the Li electrode under the electric field, and re-bond with Li ions to form a LiCl-dominated interphase layer. Analogous ion-migration behavior has been observed in a LiI-contained cathode as well as a Ag–C composite anode[61,62]. To verify this Cl-migration speculation, STEM-EDS was performed for Cl-16 after Li plating/stripping over 400 cycles (Fig. 3d). As displayed in Supplementary Fig. 14, the LiCl nanoshell is not detectable after prolonged cycling, thus supporting the speculation about Cl migration from the nanoshells to the LiCl-dominated interphase layers.

The regenerated LiCl-dominated interphase layer in turn further diminishes the decomposition of the argyrodite by the

metallic Li electrode. Such a benign cycle enables to construct gradually a dense, even, and uniform LiCl-dominated barrier layer, i.e., a self-limiting Li|SE interface. Consequently, neither agglomeration on the interface nor $Li_2S_n$ decomposition product is detected in Cl-13. In contrast, due to the lack of LiCl shells/frameworks, Cl-06 is decomposed by Li at the beginning, and the electron-conducting $Li_2S_n$ and $Li_3P$ immediately form. Therefore, a rough and irregular LiCl-dominated layer gradually forms with an uneven thickness, which is unable to prevent the continuous reactions at the interface. The less stable Li|SE interface with large pieces of agglomerated decomposition products with a size over 10 μm causes a rapidly increased interface resistance. The XRD patterns (Supplementary Fig. 15) after Li plating/stripping for 200 cycles (Supplementary Fig. 7) confirm that Cl-13 remains a larger amount of argyrodite phase than Cl-06.

The Cl-13 SE possesses the highest CCD and a moderate interface stability toward Li, while the Cl-06 and Cl-16 SEs with the smallest and highest chlorinity respectively show the lowest CCD, demonstrating that a moderate chlorinity is optimal for suppressing Li dendrites. The Li dendrite formation and growth can be understood by the synergistic effects of the LiCl shells/frameworks and the generated LiCl-dominated interphase layer. On one hand, LiCl can prevent the $Li^+$ reduction on the surface from mitigating the Li-dendrite deposit[53]. In addition, the decent bulk and shear moduli of LiCl (31 and 20 GPa) are favorable for suppressing Li dendrites[59]. Moreover, a stable interface avoids the formation of MCI, beneficial to resist the Li dendrite penetration. On the other hand, the LiCl shells/frameworks act as an electronically insulating grain boundary, which has been recently reported to be beneficial to inhibit the Li infiltration in polycrystalline SEs[63,64]. Therefore, at a moderate current density of 0.25 mA/cm$^2$, the LiCl-dominated interphase layer in the Cl-16 SE enables a stable Li plating/stripping cycling with low overpotential and thus outperforms the other SEs.

At a high current density (0.5 mA/cm$^2$), however, the Li dendrites penetrate the LiCl-dominated interphase layer and continuously grow inside the SEs. In Cl-13, the thin LiCl nanoshells are partially depleted/consumed during forming a LiCl-dominated interphase layer, the remainings allow the argyrodite core to react with the penetrated Li dendrites until they break off, which is known as the self-healing effect[65]. This process is analogous to the expansion screw effect proposed recently[24]. However, in Cl-16, because of the high chlorinity, the thick LiCl nanoshells retain in the regions far away from the Li|SE interface and overly isolate the argyrodite cores, thereby hard to consume the Li filaments as quick as the growing rate[11,24]. As a consequence, the SE with a too high chlorinity cannot withstand the high current density.

Albeit the LiCl shell is absent in Cl-06, the irregular LiCl interphase layer on the surface of Li electrode can sustain a low current density, and the reactive Li|Cl-06 feature allows the Cl-06 SE to consume the grown Li filaments[24]. Under a high current density (0.5 mA/cm$^2$), however, the interfacial degradation becomes aggravating and leads to an increased polarization voltage and a localized current density, thereby boosts the severe growth of Li filaments that far exceed the consumption. A short circuit finally occurs when these filaments are connected.

The aforementioned mechanism behind the (electro)chemical performance of argyrodite determined by chlorinity is shown schematically in Supplementary Fig. 16. In brief, a moderate chlorinity (Cl-13) gains optimal LiCl nanoshells/frameworks to enable the strongest dendrite suppression capability and the best long-term cycling stability at high current densities. The microstructure of Cl-13 meets the requirements for high performance ASSLB, i.e., it can form a(n) (electro)chemically stable LiCl-dominated interphase layer to prevent parasitic reactions and to mitigate Li dendrite growth, and meanwhile the thin LiCl nanoshells/frameworks allow to consume the Li dendrites and cast off the constraint of the interphase layer.

To understand the Li|SE interface better, we assembled Li||LNO@NCM622 ASSLBs using Cl-06 and Cl-13 SEs. The cells have been submitted to a galvanostatic charging to in situ monitor the Raman activity at the Li|SE interface. The cells were charged at 0.1 mA/cm$^2$ from an open circuit voltage to 4.3 V at 24 °C. The Raman detections were performed in the voltage range of 3–4.3 V. By using a home-made setup (Fig. 10a), the laser beam was focused on a fixed point (Fig. 10b) at the cross-section of Li|SE. The in situ Raman spectra of the cell using Cl-06 and Cl-13 SEs are respectively shown in Fig. 10c, d. The Raman spectra for both cells before cycling are similar. The highest peak at 421 cm$^{-1}$ and other peaks at 205, 272, and 573 cm$^{-1}$ are assigned to the $PS_4^{3-}$ ions in argyrodite[61], indicating a proper detection site is selected. At the beginning of the charging state, no peak change is observed. After charging to 3.6 V, for the Cl-06 cell, new peaks at 198 and 486 cm$^{-1}$ appear clearly, which are attributed to the characteristic band of $P_2S_5$ and $Li_2S_n$[66,67], respectively. The intensities of these two peaks gradually raise with increasing the charging voltage. Moreover, a shoulder in the region of 112–135 cm$^{-1}$ assigned to $P_2S_5$ arises immediately when the voltage over 4 V. Throughout the entire process, the peak intensities of the $PS_4^{3-}$ unit rapidly decrease. The decomposition products for these asymmetric cells are in well agreement with those for the symmetric cell (Figs. 8b and 9b). For the Cl-13 cell (Fig. 10d), the peak attributed to the decomposition products is not observed, and the peak intensities of $PS_4^{3-}$ show only limit decreases upon the charging process, exhibiting a self-limiting interface feature of Li|Cl-13.

In summary, we demonstrate that the chlorinity in a Li argyrodite strongly affects the (electro)chemical performance on the Li|argyrodite interface including the formation of interphase, interfacial microstructure, and Li dendrite suppression capability. A moderate Cl content (Cl = 1.3) is found to be an optimized chlorinity to gain the highest CCD and the most compatible Li|argyrodite interface upon a long-term electrochemical cycling, thereby sustaining a larger current density than the others with lower or higher Cl contents, though it only has a medium ionic conductivity (5.3 mS/cm) and a(n) (electro)chemical stability against Li. With the assistance of the cryo-STEM and EDS, the distributions of Cl atoms in Li argyrodite are found varying with the chlorinity. Instead of locating entirely on the crystal lattice analogous to the Cl-poor argyrodites as supposed, the Cl atoms in the Cl-rich argyrodites show a differential distribution that a majority exists on the grain surfaces to form LiCl nanoshells while a minority on the argyrodite lattice to substitute the S atoms. The localized argyrodite grains were wrapped into the LiCl frameworks. This peculiar microstructure facilitates the Cl ions in the frameworks to migrate to the Li|SE interface for reconstructing a dense, even, and uniform LiCl-dominated layer next to the Li electrode during electrochemical cycling. The electronically insulated LiCl with decent bulk and shear moduli acts as a self-limiting interphase to impede the (electro)chemical redox and to mitigate Li dendrites. Consequently, the decomposition of argyrodite is significantly prevented and the appearances of $Li_2S_n$ and $P_2S_5$ clusters are restrained. The cells using Cl-13 SE demonstrates a high capacity and good cycling stabilities at a current density of 0.5 mA/cm$^2$.

## Methods

**Materials synthesis**. Regarding argyrodite SE fabrication, the $Li_2S$ (aladdin, 99.9%), LiCl (aladdin, 99.9%), and $P_2S_5$ (Alf Aesar, 99.9%) powders were weighted with a total mass of 2 g according to the stoichiometric molar ratio of $Li_{7-x}PS_{6-x}Cl_x$ ($x$ = 0.6, 1.0, 1.3, 1.45, and 1.6), and mixed in an agate mortar for 15 min. The

powders were subjected to a ball milling (Fritsch P7 premium) with 10 mm diameter balls (tungsten carbide) for 1 h premixing at 100 rpm and then for 10 h milling at 550 rpm for mechanochemical synthesis. The resultant powders were pressed into pellets and vacuum sealed in a quartz tube for annealing at 450–550 °C for 12 h with a ramping rate of 5 °C/min[20,21], and then with a cooling rate of 2 °C/min to 350 °C followed by naturally cooling to 24 °C. The heat-treated pellets were hand-ground in an agate mortar for further uses. $Li_{7-x}PS_{6-x}Cl_x$ (x = 0.6, 1.0, 1.3, 1.45, and 1.6) with different x are designated as Cl-06, Cl-10, Cl-13, Cl-145, and Cl-16, respectively. The samples were protected in an argon atmosphere ($H_2O$, $O_2 \leq 1$ ppm) throughout all procedures. The details of the commercial materials are: LNO@NCM622 and LNO@NCM811 active materials (4 μm diameter, GLESI China), Li metal (99.9%, 50 μm thickness, MTI China), In (99.99%, 50 μm thickness, Hawk China), LSPSC (GLESI, China), vapor-grown carbon fibers (VGCF, 150 nm diameter, Showa Denko Japan), and Pt foil (100 μm thickness, 99.99%, Hawk China). All purchased materials were used as-received without further treatment, except that the VGCF was vacuum-dried at 110 °C for 12 h.

**Materials characterization**. XRD was carried out using a Rigaku D/MAX-2500/PC (Cu Kα) in a 2θ range of 10–70° at a scan rate of 3 °/min. The powders were sealed on a sample holder with polyimide film in an Ar glove-box ($H_2O$, $O_2 \leq 1$ ppm) to isolate from moisture. In situ Raman scattering tests were carried out using a Renishaw inVia system with a 532 nm excitation source with a home-made spectro-electrochemical cell. The cell was sealed by fluorine gaskets between cell body and transparent quartz window (10 mm in diameter), which allows the laser beam focused on the lateral surface (at interface regions, Fig. 10a) of the battery pellet. XPS was performed on a Thermo Scientific K-Alpha+. Surface and cross-section morphology were taken with a Helios g4 cx SEM equipped with EDS. TEM image and SAED were acquired in a Cs-corrected environmental transmission electron microscope (ETEM, Titan G2, Thermo Fisher Scientific) at 300 kV. The sample was transferred in an Ar-protected holder. STEM-HAADF and STEM-EDS were performed in a TEM (Themis Z, Thermo Fisher Scientific) at 300 kV with a monochromator and a spherical aberration corrector (Cs, CEOS GmbH) and an EDS detector (SuperXG21). The collection semi-angle of the STEM detectors was set to 41-200 mrad for HAADF imaging. The TEM gird was loaded onto the cryo-holder in Ar (99.999%) atmosphere and quickly transferred to the STEM in cryogenic temperature. For the ex situ measurements (XPS, SEM-EDS, XRD) of the interface, the specimens were collected from the Li|SE interface layer after detached the cycled Li||Li symmetric cells in an Ar-filled glove box ($H_2O$, $O_2 \leq 1$ ppm). The specimens were loaded into an air-tight sample holder and then transport from the Ar-filled glovebox to the equipments used for the ex situ measurements.

**Electrochemical characterization**. EIS measurements were carried out at 24 °C on a Princeton P4000 in the frequency range from 0.1 Hz to 1 MHz. The ionic conductivity was evaluated on an In|SE|In symmetric cell at 24 °C. In foil was placed on both sides of the SE pellets as blocking electrodes. The measurement error is within 3%. The CCD and Li plating/stripping galvanostatic measurements were performed on Li|SE|Li symmetric cells at 24 ± 4 °C at different current densities on a LAND cell test system. The error bar for the CCD data was disclosed by calculating the standard deviation (SD) of CCD values based on the equation: SD = $\sqrt{\frac{1}{n}\sum_{i=1}^{n}(x_i - \bar{x})^2}$, where n is the sample size (n = 3 in this work), $x_i$ is the individual CCD values, $\bar{x}$ is the mean value of the CCD data. DC polarization was carried out on Pt|SE|Pt symmetric cells at 24 °C using Princeton 4000 by applying a constant voltage of 5 mV to determine the electronic conductivity. A Faraday cage was used during the measurement. The pellets for ionic conductivity, Li|SE chemical stability (time-resolved impedance tests), CCD, Li plating/stripping cycling, and DC polarization were cold-pressed from the powders at 250 MPa. All these symmetric cells were finally mounted on an air-tight two-electrode cell. Stainless steel rods were used as current collectors for these tests. The external pressure applied during these measurements was ~3 MPa.

**All-solid-state Li metal cell assembly and testing**. ASSLBs were assembled in an Ar-filled glove box using LNO@NCM622 (or LNO@NCM811) active material and Li (50 μm) (or Li–In) electrode. The Li–In electrode (~40 μm) was prepared via manually rolling a piece of Li with a piece of In with a weight ratio of 5:95 using a stainless steel rod, and then cut into disks with a diameter of 10 mm. For preparing the positive electrodes, LNO@NCM622 (LNO@NCM811) and LSPSC ion conductors were mixed manually in a mortar for 30 min with a mass ratio of 7:3 and together with 1 wt% of VGCF electron-conductor. Bilayer pellets (10 mm in diameter) containing the positive electrode (4 mg/cm²) and $Li_{7-x}PS_{6-x}Cl_x$ (x = 0.6, 1.3) SEs (70 mg) were pressed at ~360 MPa, then Li–In or Li foil was attached to the electrolyte side. The final battery was sandwiched by stainless steel rods used as current collectors in an air-tight two-electrode cell. An external pressure of 6 MPa was applied during the cell cycling. Galvanostatic charging/discharging (Neware, CT4008) of the cells were performed at 24 ± 4 °C under cut-off voltages of 2.4–4.3 V (vs. Li/Li+) for Li electrode and 2.0–3.68 V (vs. Li–In/Li) for Li–In electrode. For in situ Raman spectroscopy measurement, the Li|SE|LNO@NCM622 battery pellet (10 mm in diameter) was pressed following above-mentioned process and then put into a home-made spectro-electrochemical cell aforementioned and cycled at 24 °C on a CT4008 under cut-off voltages of 2.4–4.3 V (vs. Li/Li+) and a

current density of 0.1 mA/cm². The battery assembling was carried out in an Ar-filled glove box ($H_2O$, $O_2 \leq 1$ ppm).

## Data availability

The data that support the findings of this study are available from the corresponding author upon reasonable request. Source data are provided with this paper.

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

## Acknowledgements
This work was supported by the National Natural Science Foundation of China (52172243) to L.Z.

## Author contributions
L.Z. and D.Z. conceived the research and designed the experiments. D.Z., J.Y., and R.X. performed the experiments. S.W., X.Y., C.Y, and L.W. participated in part of the characterization and data analysis. All authors discussed the results and wrote the manuscript.

## Competing interests
The authors declare no competing interests.
