## [Peer Review File · Nature Communications]

REVIEWER COMMENTS

Reviewer #1 (Remarks to the Author):

General comments

In the present study, the authors have investigated lithium argyrodite solid electrolytes of the general formula $\text{Li}_{7-x}\text{PS}_6\text{-xCl}_x$ (LPSCI) with different Cl content. They find that an increased Cl content, leads to an improved cycling performance with regard to a Li anode. Such observations are rationalized via a micrometer thick LiCl layer between the Li metal anode and the SE. The study is well conducted and conclusions are reasonable supported by the research data. However, a few important points need to be addressed before publication. Thus, I recommend this work to be published in Nature Communications after major revisions. Point by point comments are listed in the following:

Line 58-61: improper citation

Line 97-98: A trace of Li_2S and LiCl impurities are observed for the low ($x=0.6$) and high chlorinity ($x \geq 1.3$) sample

1) Shown in Fig.1a, on the contrast to the description, there is no impurity for Cl-13.

2) Please add the LiCl impurity content in Table S1 as derived from the refinement

Line 106-107: the authors show the SAED patterns taken from TEM, found the crystallinity difference between Cl-06 and Cl-16

1) In Figure 1 the authors show SAED patterns for Cl-06 and Cl-16, however in the subsequent manuscript and especially with regards to cell cycling they compare Cl-06 with-13. So please also show SAED pattern for Cl-13.

Line 111-112: the authors described the incomplete semicircle (Fig. 1d) as grain boundary resistance

1) It is hard to distinguish the grain boundary and bulk contribution to EIS results as there are no semicircles visible, even in the enlarged inset of Fig. 1d. The incomplete semicircle should be treated as total resistance, please adjust this in the main text.

Line 138-140: the authors stated the R_{int} enlarged more intensively for Cl-06 during the aging time (Fig. 2a and f)

1) The time-resolved EIS test result for Cl-06 shows significant differences with the others. For Cl-06, the EIS has two sections, semicircle followed the tail, while others have three parts, the incomplete semicircle, semicircle and the tail. Please rationalize the reason why the semicircle for Cl-06 standing for R_{int} , otherwise the statement, increasing Chlorinity could stable the chemical stability, is weak. Additionally, please add EIS fitting detail information in the supplement.

Fig. 4 :

1) The comparison of the cell cycling Electomechanical performances between Cl-10 and Cl-16 is not fully valid. As the authors applied a rather high current rate (1C), the difference in reversible capacity comes as a result of the different ionic conductivities of the Cl-06 and Cl-13. Please comment on this in the main text to make this point clear to the reader.

Line 356-357: the authors stated it could be some residual LiCl·H₂O due to the trace of water.

1) Improper rationalization and citation. It is most likely impossible that LiCl·H₂O under high temperature annealing does decompose. If the authors insist in this statement please provide further experimental evidence such as IR spectroscopy to detect possible OH vibrations related to H₂O crystal water.

Line 467: mechanochemical instead of "echanochemical"

Line 473: Cu K α instead of K α

Fig. 6. The authors say that a ~ 55 μm thick LiCl layer forms between the SE and the Li metal anode. In Fig. 6 however indicates that it is more likely to be a mixture of LPSCI and LiCl,

which would also explain the reasonable charge transfer kinetics of such layer, as pure LiCl is a poor ion conductor. Please soften the conclusive statements. It might be better to talk about a LiCl-rich interface layer. Please consider this in the main text.

Conclusion section: Please comment about the possibility that a LiCl interface layer prevents Li dendrite formation just because of the hardness (mechanical properties) of LiCl versus LPSCI

Improve experimental section:

1) in particular give the external pressure applied during the electrochemical measurements (EIS, battery cycling)

2) Which electrodes have been used for the EIS measurements

Fig. 7. The authors state that LiCl form a homogeneous surface layer on the LPSCI particles. However from Fig. 7 it seem like more a secondary impurity phase in between the LPSCI particles. Please comment on this.

Reviewer #2 (Remarks to the Author):

The author interprets LiCl layer formation interestingly in cell performance due to the influence of Cl contents in argyrodite.

Similar research papers regarding LiCl layer formation and Cl-rich doping effects through interfacial reactions at Li metal of argyrodite (S. Wenzel et al. *Solid State Ionics* 318 102-112 (2018) & W. Arnold et al. *J. Power Sources* 464, 228158 (2020)) have already been reported. It was new to observe the formation of LiCl nanoshell through a new method of Cryo-STEM analysis, but it would be better to publish this paper in a more specific journal by referring to the questions and comments below.

1)

In general, when the substitution amount of Cl increases, some cannot react and exist as impurities such as LiCl. This unreacted LiCl causes a decrease in ion conductivity, but an increase in irregularity and Li vacancy generation due to the substitution of Cl offset this decrease.

If this is LiCl nano-shell as the author claims, the author needs to explain how LiCl nanoshell was created compared to previous reports. Or, what process difference made LiCl nano-shell formation possible?

2)

In addition, it is questionable that the composition of Cl_{0.6} was set as a reference. The author would have compared it to see whether the LiCl layer was created, but it is difficult to compare Cl_{0.6} with high Cl contents because it is structurally unstable and different materials, not just because of the presence of the LiCl layer.

3)

In addition, at low current density, high Cl content works as the author assumes, but when the current density is increased, short occurs within tens of cycles. The explanation for this is not clear.

4)

LiCl exists as a very thin layer (50 nm) on the argyrodite surface, but how can it be observed as XRD peak? If it is detected by XRD, it exists as an impurity to some extent in my opinion. If so, Cl_{1.6} will eventually affect long-term cell performance, even if the conductivity is good.

5)

The description is made by mixing the functions of LiCl in the form of impurities existing in argyrodite synthesis and LiCl generated as a decomposition product at the Li interface. The relationship between LiCl nano-shell and cycling characteristics is unclear. LiCl of the nano-shell moves toward Li to form a stable SEI layer, which improves stability?

6)

As the author mentioned, the ion conductivity of LiCl is very low, and if this resistance layer is formed to a thickness of 50 μ m at the interface with Li, the impedance of this layer is expected to be very large and it is questionable whether a normal electrochemical reaction is possible.

RESPONSE TO REFEREES

The authors highly appreciate the editor and reviewers' activities for pointing out a series of valuable points to improve the manuscript. According to the reviewers' comments and suggestions, we carefully revised the manuscript. All parts changed in the revised manuscript are highlighted with yellow background. We hope the revisions are acceptable. Thank you again for your time and kind help.

Comment of Referee 1

General comments

In the present study, the authors have investigated lithium argyrodite solid electrolytes of the general formula $\text{Li}_{7-x}\text{PS}_{6-x}\text{Cl}_x$ (LPSCI) with different Cl content. They find that an increased Cl content, leads to an improved cycling performance with regard to a Li anode. Such observations are rationalized via a micrometer thick LiCl layer between the Li metal anode and the SE. The study is well conducted and conclusions are reasonable supported by the research data. However, a few important points need to be addressed before publication. Thus, I recommend this work to be published in Nature Communications after major revisions. Point by point comments are listed in the following:

Response:

We thank the reviewer for these inspiring comments and the recommendation to publish.

Line 58-61: improper citation.

Response:

We thank the reviewer for the comment. We have modified the text and re-arranged the citations. Original Ref. 15 reports the important role of site disorder and lattice softness on the ion transport, so we modified the text and added two of the initial references describing the site disorder of S/X in argyrodites (Refs. 17 and 18: Deiseroth H. et al., *Angew. Chem. Int. Ed.* 47, 755–758 (2008); Rayavarapu PR. et al., *J. Solid State Electrochem.* 16, 1807–1813 (2012)): "Among different X elements, the $\text{Cl}^-/\text{S}^{2-}$ structure is the most disordered with proper lattice softness, leading to a prominent ionic conductivity in $\text{Li}_6\text{PS}_5\text{Cl}$ ^{16, 17, 18}."

Original Ref. 16 is a review article regarding argyrodites. We moved it to somewhere else and replaced with the first experimental paper for $\text{Li}_{7-\alpha}\text{PS}_{6-\alpha}\text{Cl}_\alpha$ ($\alpha > 1$) argyrodites (Ref. 19: Adeli, P. et al. *Angew. Chem. Int. Ed.* 58, 8681–8686 (2019)).

Original Ref. 17 reports the synergistic effect of site disorder and lattice softness on the ion transport for $\text{Li}_{7-\alpha}\text{PS}_{6-\alpha}\text{ClBr}_\alpha$ ($0 \leq \alpha \leq 0.8$) through combining solid-state NMR, neutron scattering analysis, and theoretical calculation/simulation. Although it is not wholly talking about Cl substitution but partially with Br substitution, we think the result is interesting and important and thus cite it here.

Original Refs. 18-19 report the ion transport in $\text{Li}_{7-\alpha}\text{PS}_{6-\alpha}\text{Cl}_\alpha$ ($\alpha > 1$) argyrodites. In addition, we added the reference (Ref. 22: de Klerk, NJJ. et al. *Chem. Mater.* 28, 7955–7963 (2016)) that first suggests the way theoretically to improve the ionic conductivity via increasing the halogen concentration and optimize their distribution over the 4a and 4c sites on argyrodites.

Line 97-98: A trace of Li_2S and LiCl impurities are observed for the low ($x=0.6$) and high chlorinity

($x \geq 1.3$) sample

1) Shown in Fig.1a, on the contrast to the description, there is no impurity for Cl-13.

2) Please add the LiCl impurity content in Table S1 as derived from the refinement.

Response:

Q 1)

We are sorry for the unclarity for the description. In the revised version, we marked the LiCl peak on each profile shown in updated Fig. 1a. As shown, the LiCl impurity exists in the samples where $x = 1.3, 1.45, 1.6$.

Q 2)

The LiCl impurity contents derived from refinements were added in Table S1. The LiCl impurity appears for $x \geq 1.3$, but the quantity detected from XRD is very low and slightly increases with increasing Cl concentration. The LiCl impurity contents are 2.7, 3.2, and 4.4 vol.% for $x = 1.3, 1.45$, and 1.6, respectively. We incorporated this clarification in the updated manuscript (Page 4, Paragraph 1).

Additionally, we carried out SEM-EDS on a cold-pressed pellet of the pristine Cl-13 at several regions (more than 7 regions but some of them are burned by the electron beam) selected randomly under a large magnification (20,000 \times , the samples are burned by the electron beam when the magnification is higher than this). The typical images are displayed in Supplementary Fig. 9. Except one region displays an agglomerated LiCl particle ($\sim 2 \mu\text{m}$), the other regions all display a homogeneous Cl-distribution, indicating that the proportion of the agglomerated LiCl particles is very low. We incorporated this clarification in the updated manuscript (Page 15, Paragraph 1).

Additionally, to soften some statements with a more precise description, we removed the sentence at Line 323-326.

Supplementary Figure 9. SEM-EDS images of the pristine CI-13 pellet tested more than 7 regions (20,000×). **a**, Selected images on the region with homogeneous elemental distribution. **b**, The only region with an agglomerated LiCl particle.

Line 106-107: the authors show the SAED patterns taken from TEM, found the crystallinity difference between CI-06 and CI-16

1) In Figure 1 the authors show SAED patterns for CI-06 and CI-16, however in the subsequent manuscript and especially with regards to cell cycling they compare CI-06 with-13. So please also show SAED pattern for CI-13.

Response:

We showed the SAED patterns of CI-06 and CI-16 for comparison because they are the two end points. Followed the good suggestion we also show the TEM image and the SAED pattern for CI-13 as well as CI-10. CI-13 was added in Fig. 1c and CI-10 was shown in Supplementary Fig. 1 in the revised version. CI-16 was also replaced by new SAED images due to the original one was off zone axis. The amorphous halos are marginally seen in CI-13 and CI-16, but not seen in CI-06 and CI-10. The reason for the trace of amorphous phase is still unknown. We rearranged the figures and modified the text in the updated manuscript (Page 4, Paragraph 1): “The selected area electron diffraction (SAED) patterns (Fig. 1c and Supplementary Fig. 1) taken from TEM show that CI-06 and CI-10 are well crystallized, whereas CI-13 and CI-16 marginally show a small amount of an amorphous phase as indicated by the weak halos. This is obviously against the expectation for some not yet known reasons, as the argyrodite phase should be in a crystalline state after annealing²⁷” and deleted the

amorphous speculation on Line 329-333 of the original manuscript.

Fig. 1c TEM images of CI-06, CI-13, CI-16 and the corresponding SAED patterns.

Supplementary Figure 1. TEM image of CI-10 and the corresponding SAED pattern.

Line 111-112: the authors described the incomplete semicircle (Fig. 1d) as grain boundary resistance 1) It is hard to distinguish the grain boundary and bulk contribution to EIS results as there are no semicircles visible, even in the enlarged inset of Fig. 1d. The incomplete semicircle should be treated as total resistance, please adjust this in the main text.

Response:

This is a very constructive comment. In view of the small grain boundary resistance regarding sulfide solid electrolytes (Dawson J.A. et al., Chem. Mater. 31, 5296–5304 (2019)), we should attribute the incomplete semicircle to the total contribution of bulk and grain boundary. Thanks for pointing out the inappropriate description. We adjust the main text in the revised manuscript (last paragraph in

Page 4): “it is hard to distinguish the grain boundary and bulk contribution based on these measured impedance spectra. With the comparable micro-level morphology observed from the SEM images (Supplementary Fig. 2) for the cold-pressed Cl-06 and Cl-13 SE pellets, the steadily shrunken semicircle with increasing Cl concentration can be attributed to the reduced grain-boundary/bulk resistance” and the references (Refs. 28 and 29: Dawson J.A. et al., Chem. Mater. 31, 5296–5304 (2019); Kamaya N. et al., Nat. Mater. 10, 682–686 (2011)) were cited to support the statement of “The incomplete semicircles indicate a small grain boundary resistance” (last paragraph in Page 4). We also modified the text in paragraph 1 in Page 11: “First, a middle chlorinity (Cl-13) exhibits the highest CCD, while surprisingly the highest chlorinity (Cl-16) shows the lowest CCD, despite its highest ionic conductivity.”

Line 138-140: the authors stated the R_{int} enlarged more intensively for Cl-06 during the aging time (Fig. 2a and f)

1) The time-resolved EIS test result for Cl-06 shows significant differences with the others. For Cl-06, the EIS has two sections, semicircle followed the tail, while others have three parts, the incomplete semicircle, semicircle and the tail. Please rationalize the reason why the semicircle for Cl-06 standing for R_{int} , otherwise the statement, increasing Chlorinity could stable the chemical stability, is weak. Additionally, please add EIS fitting detail information in the supplement.

Response:

We thank the reviewer for the comment. The EIS curve of Cl-06 at 0 hour is also a semicircle followed the tail, as shown in the figure below (Fig. R1). However, the interfacial resistance intensively increases with increasing time and thus gradually overlaps the grain and grain boundary parts. The EIS fitting curves and the corresponding R and RQ values for the selected time at 300 h were added and shown in Supplementary Fig. 3. The C values for the interface contribution are within 10^{-6} F, in agreement with the characteristic capacitance of interface part. Moreover, the start and end frequency values for the R_{int} region were added in Fig. 2a-d, which visually illustrate the differences in R_{int} change.

Fig. R1 The EIS curve of Cl-06 at 0 hour.

Fig. 2a-d Impedance spectra for the time-resolved EIS tests.

Supplementary Figure 3. Selected fitting result of time-resolved EIS spectra on the $\text{Li}|\text{Li}_{7-x}\text{PS}_{6-x}\text{Cl}_x|\text{Li}$ symmetric cells aging for 300 h.

Fig. 4 :

1) The comparison of the cell cycling Electromechanical performances between CI-10 and CI-16 is not fully valid. As the authors applied a rather high current rate (1C), the difference in reversible capacity comes as a result of the different ionic conductivities of the CI-06 and CI-13. Please comment on this in the main text to make this point clear to the reader.

Response:

We thank the reviewer for the comment. In view of the same cathode and anode materials on a cell, the reversible capacity may depend on the ionic conductivity of SE and the SE/anode interface. In this work, although CI-06 shows a lower ionic conductivity than CI-13 and CI-16, the value of CI-06 is still over 1 mS/cm that is generally considered enough for working at a moderate current density. Therefore, in this case, an inferior SE/anode interface could be the main factor to degrade the cell capacity due to the high interfacial resistance. We tested the LNO@NCM811|CI-16|In-Li cell at 1C. It shows nearly the same capacity as CI-13, only about 3 mAh/g lower. Its charging/discharging performance was added in Supplementary Fig. 5 in the revised manuscript. Consequently, we think the difference in reversible capacity mainly originates from the different interfacial behaviors of SE against deposited Li. A clarification was added at the last paragraph in Page 10: "the specific capacity of CI-13 (~148 mAh/g@50th cycle) is higher than that of CI-06 (~123 mAh/g@50th cycle) and is comparable to that of CI-16 (~145 mAh/g@50th cycle, Supplementary Fig. 5). This is because CI-06 consumes more active Li than CI-13 and CI-16 at the SE/Li interface..."

Additionally, we split Fig. 4 into two figures (Figs. 5 and 6) to improve the accessibility and readability. We also moved the charging/discharging profiles of LNO@NCM811|In-Li cells using CI-06 and CI-13 SEs from Supplementary Fig. 3 (original version) to Fig. 6b, c. The main text has been updated correspondingly.

Supplementary Figure 5. Charge/discharge profiles (a) and capacity and Coulombic efficiency (b) of the LNO@NCM811|CI-16|Li-in cell run at 1C.

Line 356-357: the authors stated it could be some residual LiCl·H₂O due to the trace of water.

1) Improper rationalization and citation. It is most likely impossible that LiCl·H₂O under high temperature annealing does decompose. If the authors insist in this statement please provide further experimental evidence such as IR spectroscopy to detect possible OH vibrations related to H₂O crystal water.

Response:

We thank the reviewer very much for this important suggestion. To clarify the speculation, an accurate DSC test was performed on a Perkin-Elmer DSC8000 for CI-16 powders from 50–300 °C, as shown in the figure below (Fig. R2). No any thermal peak is observed except the background. So,

there is no residual $\text{LiCl}\cdot x\text{H}_2\text{O}$ in the sample. We thus removed this speculation.

Fig. R2 Differential scanning calorimetry (DSC) curve of the Cl-13 powders at a heating rate of $10^\circ\text{C}/\text{min}$ in the range of $50\text{--}300^\circ\text{C}$. No any thermal peak is observed.

On the other hand, we believe the space charge effect is one of the reasons to enable an increased ionic conductivity with increasing chlorinity. This effect widely exists in composites such as the conductor-insulator and conductor-conductor systems (J. Maier, *Nat. Mater.* 4, 805–815 (2005); W. Arnold et al. *J. Power Sources* 464, 228158 (2020)). In this, a high conduction matrix is the crucial factor for the overall conductivity of a composite (E. Rangasamy et al., *J. Mater. Chem. A* 2, 4111–4116 (2014)). Cl-rich argyrodites with high ionic conductivity due to the enhanced lattice disorder and Li vacancies are thus a proper matrix. Therefore, LiCl nano-shells do not degrade the ion transport of the argyrodite-LiCl composite. However, the increase of ionic conductivity is not the main concern in this work. Inspired by this work, deeper insights on ion transport behavior in argyrodite may appear in future.

Clarification added (Page 17, Paragraph 1): “In addition, we speculate that a space-charge layer exists between argyrodite and LiCl nanoshell, which also contributes to an increased ionic conductivity. The space charge effect widely exists in composites such as the conductor-insulator and conductor-conductor systems^{51,52}. In this, a high conduction matrix is the crucial factor for the high overall conductivity of a composite⁵³. The Cl-rich argyrodite with a high ionic conductivity due to the enhanced lattice disorder and Li vacancies belongs thus to such a proper matrix. Therefore, the LiCl nanoshell does not degrade the ion transport of the argyrodite-LiCl composite SEs.”

Line 467: mechanochemical instead of “echanochemical”.

Response:

We thank the reviewer for pointing out to this typo. We corrected this in the revised version of the manuscript.

Line 473: Cu K α instead of Ka.

Response:

Many thanks. It was corrected.

Fig. 6. The authors say that a $\sim 55 \mu\text{m}$ thick LiCl layer forms between the SE and the Li metal anode. In Fig. 6 however indicates that it is more likely to be a mixture of LPSCI and LiCl, which would also explain the reasonable charge transfer kinetics of such layer, as pure LiCl is a poor ion conductor. Please soften the conclusive statements. It might be better to talk about a LiCl-rich interface layer. Please consider this in the main text.

Response:

Thank you very much for this very constructive comment. You are right. Besides Cl, P and S also exist in the boundary layer as observed from the EDS mapping (Fig. 8c). Considering that the LiCl layer comes from the LiCl nanoshells and the XPS analysis on the SE/Li interface region (Fig. 7) reveals a large proportion of PS_4^{3-} , the boundary layer should contain argyrodite and/or Li_3PS_4 in addition to LiCl, i.e., a LiCl-dominated interface layer mixed with Li superionic conductor. In literature it has been reported that a solid electrolyte interface (SEI) with a nanosized LiCl mixture possesses fast charge-transfer kinetics (Refs 46 and 47: Q. Zhao, et al., *Angew. Chem. Int. Ed.* 57, 992–996 (2018); Z. Li, et al., *Energy Storage Mater.* 45, 40–47 (2022)). Furthermore, such an in-situ formed LiCl interface layer is dense and tightly contacted with Li anode, thereby conducive to gain low resistance SEI, which has been found on a LiF-rich SEI layer (Ref. 48: X. Fan, et al., *Sci. Adv.* 4, eaau9245 (2018)). As a result, the LiCl-dominated interface layer with a thickness of $\sim 55 \mu\text{m}$ allows fast ion transport kinetics, which is verified through the low overpotentials seen from the symmetric cells performing Li plating/stripping for hundreds of cycles (Fig. 3d, e and Fig. 4a). Based on this clarification, we modified the text and added the references in the updated manuscript (Page 14, Paragraph 1): "...a LiCl-dominated interphase layer is found next to the Li electrode. Considering that this LiCl layer comes from the LiCl nanoshells (see below) and a large proportion of PS_4^{3-} exists on the SE/Li interface region (Fig. 7), this LiCl layer should contain argyrodite and/or Li_3PS_4 in addition to LiCl, i.e., a LiCl-dominated interphase layer mixed with Li superionic conductor. This layer is smooth and homogeneous with a thickness of $55 \mu\text{m}$, where the particles must be small because no agglomeration is observed. It has been reported that a solid electrolyte interface (SEI) with a nanosized LiCl mixture possesses fast charge-transfer kinetics^{46,47}. From the SEM image (Fig. 8c), we know that this is a high quality LiCl-dominated interphase layer as it is dense, even, and uniform, which is favorable to gain a low resistance⁴⁸. Therefore, even though the LiCl interphase layer is as thick as $55 \mu\text{m}$, it still allows fast ion transport kinetics. LiCl is also an excellent electron insulator, so a high quality LiCl-dominated interphase layer is a good buffer layer to stabilize the SE/Li interface. On the contrary, the LiCl layer in Cl-06 is irregular, showing a rough edge and uneven thicknesses in a range from few to $50 \mu\text{m}$. This well interprets why the Cl-rich argyrodite SEs can easily keep constant and low overpotentials (Fig. 3d, e and Fig. 4a) during the Li plating/stripping process, i.e., via forming a self-limit interface".

We replaced pure LiCl with LiCl-dominated throughout the updated manuscript.

Conclusion section: Please comment about the possibility that a LiCl interface layer prevents Li dendrite formation just because of the hardness (mechanical properties) of LiCl versus LPSCI.

Response:

We thank the reviewer for the suggestion. Larger bulk and shear moduli are generally related to a higher hardness/strength, which may be favourable for mitigating the Li dendrite formation. We added the clarification and corresponding references (Ref. 56: J. Wang, et al., *Mater. Chem. Phys.* 244, 122733 (2020)) in the Discussion and Conclusion sections in the updated manuscript (last paragraph

in Page 18): “In addition, the decent bulk and shear moduli of LiCl (31 and 20 GPa) are favourable for suppressing Li dendrites⁵⁶”, and “...with decent bulk and shear moduli...” (Conclusion section in Page 22).

Improve experimental section:

1) in particular give the external pressure applied during the electrochemical measurements (EIS, battery cycling)

2) Which electrodes have been used for the EIS measurements.

Response:

Thanks for the reminding. More detailed information was added in Methods section in the updated manuscript (Page 23):

“Indium foils were used as blocking electrodes.”

“...ionic conductivity, SE/Li chemical stability (aging time), CCD, Li plating/stripping cycling ... Stainless steel rods were used as current collectors for these tests. The external pressure applied during these measurements was ~3 MPa.”

“An external pressure of 6 MPa was applied during the cell cycling.”

Fig. 7. The authors state that LiCl form a homogeneous surface layer on the LPSCI particles. However from Fig. 7 it seem like more a secondary impurity phase in between the LPSCI particles. Please comment on this.

Response:

We thank the reviewer for the reminding. We modified the drawing in the revised manuscript.

Fig. 9 Schematic depicting the advantages of the Cl-rich (Cl-13) compared with the Cl-poor (Cl-06) argyrodites and the formation of the differentiated interface of SE/Li.

Comment of Referee 2

The author interprets LiCl layer formation interestingly in cell performance due to the influence of Cl contents in argyrodite.

Similar research papers regarding LiCl layer formation and Cl-rich doping effects through interfacial reactions at Li metal of argyrodite (S. Wenzel et al. *Solid State Ionics* 318 102-112 (2018) & W. Arnold et al. *J. Power Sources* 464, 228158 (2020)) have already been reported. It was new to observe the formation of LiCl nanoshell through a new method of Cryo-STEM analysis, but it would be better to publish this paper in a more specific journal by referring to the questions and comments below.

Response:

We thank the reviewer for raising the comments/concerns. This we believe is caused by some misunderstandings that mainly suggest we need to improve our presentation e.g., on the existence of LiCl shells, the relation between LiCl shell and LiCl interphase layer, as well as to clarify the novelty of the present work. Please thus allow us to make the clarification. We hope the reviewer can reconsider the publication of the revised version in this journal.

Although there is a mass of papers, and still with a rising trend, on Li argyrodite solid electrolytes (SEs) regarding the chemical structure, the ion transport, the interface stability toward electrodes, and the cycling performance, so far, NO work discovers such a unique microstructure regarding halogen distribution in the argyrodites, which is found for achieving a stable SE/Li interface and a good Li dendrite suppression capability simultaneously. Here we disclosed/highlighted that the halogen distribution in argyrodites is exactly critical for high battery performance. We reveal that a well-designed composition (Cl-13) can form unique LiCl frameworks with proper thickness wrapping on argyrodite subparticles/grains, acting as a crucial role on improving both the SE/Li interfacial stability and Li dendrite suppression capability. On one aspect, Cl ions migrate from the LiCl frameworks to the SE/Li interface for reconstructing a dense, even, and uniform LiCl-dominated layer, which is a key factor to improve the interfacial stability of SE/Li and is benefit for a homogeneous Li deposit. On the other aspect, a proper thickness of LiCl frameworks is a key parameter to suppress the growth of Li dendrites. In brief, the main point of our work is to illustrate that a proper microstructure in argyrodite SEs never seen before can realize both a stable SE/Li interface and a good Li dendrite suppression capability simultaneously. In parallel, we clarify the origin and mechanism behind.

Our work thus sheds the light on a new way for mitigating the interface and dendrite issues on all-solid-state batteries, as well as gives a good example on highlighting the significance of a microstructural design toward high performance sulfide SEs.

Liter. #1 (S. Wenzel et al. *Solid State Ionics* 318 102-112 (2018)) is a pioneer and important paper on interfacial components for argyrodite solid electrolytes, which is also cited in this manuscript. The authors revealed that LiX together with Li₂S, Li₃P interphases compose the interfacial constituent. This result is also included in 'Introduction' in our original manuscript. However, NO data (including other papers on argyrodite/Li interface) hints the LiX decomposition product exists as a high quality LiX-dominated layer at the interface boundary of SE/Li, which can serve as a self-limit layer to stabilize the SE/Li interface. In our manuscript, we for the first time demonstrated visually an independent LiCl-dominated layer and revealed that its morphology varies with the chlorinity and, more importantly, further illuminated the mechanism behind why it forms differentiated LiCl

interphase layer upon chlorinity, eventually leading to different interface performance and Li dendrite suppression capability.

In liter. #2 (W. Arnold et al. J. Power Sources 464, 228158 (2020)), the mole fractions of Cl are set to Cl = 1, 2, 3. We cited this reference in the updated manuscript (Ref. 52). The excess LiCl cannot properly enter the $\text{Li}_6\text{PS}_5\text{Cl}$ structure because a saturated mole ratio of S (S = 5) and Li (Li = 6) is designed for the chemical formula $\text{Li}_6\text{PS}_5\text{Cl}\cdot x\text{LiCl}$. Thus, the amount of LiCl impurity is high. In that case, the LiCl impurity basically exists in agglomerated particles but not homogeneous coating. Thus, the symmetric cell of $\text{Li}_6\text{PS}_5\text{Cl}\cdot\text{LiCl}$ SE suffers a short circuit less than 120 cycles cycling below 0.1 mA/cm². No matter the design concept or the main idea behind is different from our work.

(1) In general, when the substitution amount of Cl increases, some cannot react and exist as impurities such as LiCl. This unreacted LiCl causes a decrease in ion conductivity, but an increase in irregularity and Li vacancy generation due to the substitution of Cl offset this decrease.

If this is LiCl nano-shell as the author claims, the author needs to explain how LiCl nanoshell was created compared to previous reports. Or, what process difference made LiCl nano-shell formation possible?

Response:

We thank the reviewer very much for this constructive comment.

The prerequisites for the formation of LiCl nanoshells/frameworks include composition and sintering technique. The composition should be on a Cl-rich state, i.e., $x > 1.0$ for $\text{Li}_{7-x}\text{PS}_{6-x}\text{Cl}_x$. In parallel, heat treatment should be performed with a slow cooling rate. To evaluate the influence of the heat treatment, two different cooling processes were adopted after sintering via: 1) cooling the sample to 400 °C slowly and then quenching in the ice water, and 2) directly quenching in the ice water. As shown in Supplementary Fig. 11, there is no LiCl nanoshell for the directly quenched sample, indicating that a slow cooling process at the high temperature regions is a crucial factor. In view of the sintering effect, the LiCl nanoshells could generate through in situ precipitating from the $\text{Li}_{7-x}\text{PS}_{6-x}\text{Cl}_x$ solid solution, otherwise they would exist as agglomerated particles rather than homogeneous coating. We incorporated this clarification in the updated manuscript (last paragraph in Page 15) and in the Methodes section (Page 22).

Supplementary Figure 11. STEM-HAADF and EDS images of Cl-16 prepared with different cooling processes by cooling to 400 °C and then quenching in ice water (a), and directly quenching in ice water (b).

(2) In addition, it is questionable that the composition of Cl0.6 was set as a reference. The author would have compared it to see whether the LiCl layer was created, but it is difficult to compare Cl0.6 with high Cl contents because it is structurally unstable and different materials, not just because of the presence of the LiCl layer

Response:

We thank the reviewer for the comment. As shown in Figs.3 and 4, the critical current density (CCD) and long-term Li plating/stripping were evaluated for all samples with various Cl contents (Cl = 0.6, 1.0, 1.3, 1.6). The results revealed that Cl = 1.3 has the optimized overall performance for a good SE/Li interface stability and a Li dendrite suppression capability. This is the reason we compare the sample where Cl = 1.3 with that where Cl = 0.6.

We tested the LNO@NCM811|Cl-16|In-Li cell at 1C. Its charging/discharging performance was added in Supplementary Fig. 5 in the revised manuscript. It shows nearly the same capacity as Cl-13, but only about 3 mAh/g lower. All the three LNO@NCM811||In-Li cells using Cl-06, Cl-13, and Cl-16 SEs, respectively, have comparable charging/discharging profiles (Fig. 6 and Supplementary Fig. 5), though Cl-06 has a lower reversible capacity than Cl-13 and Cl-16. Considering that these cells have the same cathode and anode materials and the three SEs have a high ionic conductivity over 1 mS/cm that is sufficient for working at a moderate current density, the different reversible capacity can be attributed to the different interfacial resistances at the SE/In-Li interface. Despite the lower capacity, the cell using Cl-06 SE maintains good cycling stability for over 1000 cycles, indicating that Cl-06 is structurally stable with In-Li electrode. The $\text{Li}_{7-x}\text{PS}_{6-x}\text{Cl}_x$ compounds are cubic when $x > 0.25$ (A. Gautam et al., ACS Appl. Energy Mater. 4, 7309–7315, (2021)). Therefore, a stable SE/Li interface enabled by LiCl-dominated layer is responsible for the high reversible capacity of LNO@NCM811||In-Li cells. We accordingly modified the text in the updated manuscript (last

paragraph in Page 10):" the specific capacity of Cl-13 (~148 mAh/g@50th cycle) is higher than that of Cl-06 (~123 mAh/g@50th cycle) and is comparable to that of Cl-16 (~145 mAh/g@50th cycle, Supplementary Fig. 5). This is because Cl-06 consumes more active Li than Cl-13 and Cl-16 at the SE/Li interface..."

Additionally, we split Fig. 4 into two figures (Figs. 5 and 6) to improve the accessibility and readability. We also moved the charging/discharging profiles of LNO@NCM811|In-Li cells using Cl-06 and Cl-13 SEs from Supplementary figure to Fig.6b, c. The main text has been updated correspondingly.

Supplementary Figure 5. Charge/discharge profiles (a) and capacity and Coulombic efficiency (b) of the LNO@NCM811|Cl-16|Li-in cell run at 1C.

(3) In addition, at low current density, high Cl content works as the author assumes, but when the current density is increased, short occurs within tens of cycles. The explanation for this is not clear

Response:

We thank the reviewer for raising the discussion.

To improve the accessibility and readability of the data, we split Fig. 3 into two figures (Figs. 3 and 4). Fig. 4 shows the long-term Li plating/stripping reversibility tested at 0.5 mA/cm² for different Cl contents (Cl = 0.6, 1.0, 1.3, 1.6). The Cl-13 symmetric cell was well working over 1000 cycles, so we replaced the cycling curve of Cl-13 for a longer time.

Our data exhibit that the (electro)chemical behaviors against Li electrode and Li dendrite suppression capabilities vary with chlorinity. Figs. 2 and 3a show that "the best interface chemical stability and Li dendrite suppression capability appear at different chlorinities". That is, the higher the chlorine content, the more stable the SE/Li interface, while the CCD shows a parabolic relation with chlorinity. The highest CCD appears at a moderate chlorinity (Cl = 1.3) and the lowest CCD appears at a highest chlorinity (Cl = 1.6). As we mentioned in the manuscript, these results were surprising to us, so we performed diverse methods to characterize the SE/Li interface. We found different components and morphologies on SE/Li interface upon different chlorinities, especially the formation of a dense, even, and uniform LiCl-dominated interphase layer for the high chlorinity argyrodites. To investigate the origin of this unusual interphase layer, we further performed cryo-STEM tests and revealed that higher chlorinities tends to form a thicker LiCl nanoshell on argyrodite subparticles/grains.

The LiCl nanoshells mitigate the parasitic reactions of argyrodite SE at the beginning when chemically contacting and electrochemical cycling toward Li, which acts as an initial protection. During the further electrochemical cycling, the Cl ions in the nanoshells are prone to move to the surface of the Li electrode under the electric field, and re-bond with Li ions to form a LiCl-dominated interphase on

the Li surface. The regenerated LiCl-dominated interphase layer in turn further diminishes the decomposition of the argyrodite by the metallic Li electrode. Such a benign cycle enables to construct gradually a dense, even, and uniform LiCl-dominated barrier layer, which we believe is the key factor to gain an ultra-stable SE/Li interface, i.e., a self-limit SE/Li interface.

An ultrastable interface can well inhibit the interface decomposition/degradation, avoiding the formation of mixed ion–electron conducting interphases. From this perspective, the high quality LiCl-dominated interphase layer for Cl-16 SE is beneficial to resist the Li dendrite penetration at a moderate current density of 0.25 mA/cm², thus leads to a stable Li plating/stripping cycling with low overpotential outperforming other SEs. At a high current density, however, Li dendrites can penetrate the LiCl-dominated interphase layer and continually grow inside the Cl-16 electrolyte. Albeit numerous LiCl nanoshells are consumed during the formation of LiCl-dominated layer, the remaining LiCl nanoshells of Cl-16 still can obstruct the reaction of argyrodite with Li dendrites/filaments. The continuous dendrite growth causes a quick short circuit at 0.5 mA/cm². In contrast, the thin LiCl nanoshells of Cl-13 are partially depleted/consumed during the formation of LiCl-dominated interphase layer, the remaining thin LiCl nanoshells allow the argyrodite core to react with the penetrated Li dendrites until they break off, which is known as the self-healing effect (Ref. 59: C. Zheng et al., *Small* 17, 2101326 (2021)). This process is analogous to the expansion screw effect proposed recently (Ref. 60: L. Ye et al., *Nature* 593, 218-222 (2021)). Therefore, the Li dendrites are intensively mitigated by the LiCl-dominated layer together with the argyrodite cores, achieving a good dendrite suppression capability at 0.5 mA/cm².

We reorganized and modified the text in the Discussion section and added related references in the updated manuscript (last paragraph in Page 18; Page 19, Paragraph 1).

(4) LiCl exists as a very thin layer (50 nm) on the argyrodite surface, but how can it be observed as XRD peak? If it is detected by XRD, it exists as an impurity to some extent in my opinion. If so, Cl1.6 will eventually affect long-term cell performance, even if the conductivity is good.

Response:

We thank the reviewer for the comment. LiCl exists indeed as an impurity phase, only the microstructure of nanoshells is very different from the other microstructural forms such as agglomerated particles. 50 nm thickness is for each particle. By collective effect, the large number of particles accumulates LiCl with a value of ~5 vol% (obtained from Rietveld refinement to fit the XRD pattern, see Table S1 in the revised manuscript), which could be the reason for the observed XRD peak.

To confirm the microstructure, we further carried out SEM-EDS on a cold-pressed pellet of the pristine Cl-13 at several regions (more than 7 regions but some are burned by the electron beam) selected randomly under a large magnification (20,000×, the samples are burned by the electron beam when the magnification is higher than this). The typical images are displayed in Supplementary Fig. 9. Except one region displays an agglomerated LiCl particle (~2 μm), the other regions all display a homogeneous Cl-distribution, indicating that the proportion of the agglomerated LiCl particles are very low. As indicated in the literature (Ref. 52: W. Arnold et al. *J. Power Sources* 464, 228158 (2020)), the LiCl impurity is beneficial for long-term cell performance. We incorporated this clarification in the updated manuscript (Page 15, Paragraph 1).

Supplementary Figure 9. SEM-EDS images of the pristine Cl-13 pellet tested more than 7 regions (20,000 \times). **a**, Selected images on the region with homogeneous elemental distribution. **b**, The only region with an agglomerated LiCl particle.

(5) The description is made by mixing the functions of LiCl in the form of impurities existing in argyrodite synthesis and LiCl generated as a decomposition product at the Li interface. The relationship between LiCl nano-shell and cycling characteristics is unclear. LiCl of the nano-shell moves toward Li to form a stable SEI layer, which improves stability?

Response:

We thank the reviewer for the comment. At the beginning of the chemically contacting and electrochemical cycling, the LiCl nanoshells act as initial protections by inhibiting the parasitic reactions between the argyrodite SE and Li. During the further electrochemical cycling, the Cl ions in the nanoshells migrate to the surface of the Li electrode under the electric field, and re-bond with Li ions to form a LiCl-dominated interphase layer. To verify this Cl-migration hypothesis, STEM-EDS was performed for Cl-16 after Li plating/stripping over 400 cycles. As displayed in Supplementary Fig. 12, the LiCl nanoshell is gone, well confirming the Cl transformation from the nanoshells to the LiCl-dominated interphase layers.

The regenerated LiCl-dominated interphase layer in turn further diminishes the decomposition of the argyrodite by the metallic Li electrode. Such a benign cycle enables to construct gradually a dense, even, and uniform LiCl-dominated barrier layer, which we believe is the key factor to gain an ultra-stable SE/Li interface, i.e., a self-limit SE/Li interface.

Based on this discussion, we modified the text and added the figures in the revised manuscript (last paragraph in Page 17; first paragraph in Page 18; Supplementary Fig. 12)

Supplementary Figure 12. STEM-EDS of Cl-16 after Li plating/stripping for 400 cycles.

(6) As the author mentioned, the ion conductivity of LiCl is very low, and if this resistance layer is formed to a thickness of 50μm at the interface with Li, the impedance of this layer is expected to be very large and it is questionable whether a normal electrochemical reaction is possible.

Response:

We thank the reviewer for the comment. You are right that LiCl may be not an ideal ion conductor, so the LiCl framework would hinder the interparticle ion transport. In reality, an increased ionic conductivity upon increasing Cl content is observed. We propose the following reasons and cited related references (Page 17, Paragraph 1; Refs. 47, 51-53): “The major reason is that the degree of X⁻/S²⁻ structural disorder and the Li vacancy concentration increase with increasing chlorinity, which remarkably enhances the ionic conductivity. Meanwhile, the LiCl nano-shells are thin enough to allow a fast Li⁺ transport⁴⁷. In addition, we speculate that a space-charge layer exists between argyrodite and LiCl nanoshell, which also contributes to an increased ionic conductivity. The space charge effect widely exists in composites such as the conductor-insulator and conductor-conductor systems^{51, 52}. In this, a high conduction matrix is the crucial factor for the high overall conductivity of a composite⁵³. The Cl-rich argyrodite with a high ionic conductivity due to the enhanced lattice disorder and Li vacancies belongs thus to such a proper matrix. Therefore, the LiCl nanoshell does not degrade the ion transport of the argyrodite-LiCl composite SEs.”

REVIEWERS' COMMENTS

Reviewer #1 (Remarks to the Author):

All comments by the reviewers have been addressed in detail, thus the manuscript can be accepted for publication.

RESPONSE TO REFEREES

Reviewer #1 (Remarks to the Author)

All comments by the reviewers have been addressed in detail, thus the manuscript can be accepted for publication.

Response:

We thank the reviewer for the “acceptance” recommendation.